**EMBO** *reports*

# Activity-dependent extracellular proteolytic cascade cleaves the ECM component brevican to promote structural plasticity

Jeet Bahadur Singh [ID][1,2,7], Bartomeu Perelló-Amorós [ID][3], Jenny Schneeberg[4], Hadi Mirzapourdelavar [ID][4], Constanze I Seidenbecher [ID][1,5], Anna Fejtová [ID][2], Alexander Dityatev [ID][4,5,6] & Renato Frischknecht [ID][1,3✉]

## Abstract

The brain's perineuronal extracellular matrix (ECM) is a crucial factor in maintaining the stability of mature brain circuitry. However, how activity-induced synaptic plasticity is achieved in the adult brain with a dense ECM is unclear. We hypothesized that neuronal activity induces cleavage of ECM, creating conditions for synaptic rearrangements. To test this hypothesis, we investigated neuronal activity-dependent proteolytic cleavage of brevican, a prototypical ECM proteoglycan, and the importance of this process for functional and structural synaptic plasticity in the rat hippocampus ex vivo. Our findings reveal that chemical long-term potentiation (cLTP) triggers rapid brevican cleavage in perisynaptic regions through the activation of an extracellular proteolytic cascade involving proprotein convertases and ADAMTS-4 and ADAMTS-5. This process requires NMDA receptor activation and involves astrocytes. Interfering with cLTP-induced brevican cleavage prevents the formation of new dendritic protrusions in CA1 but does not impact LTP induction by theta-burst stimulation of CA3-CA1 synapses. Our data reveal a mechanism of activity-dependent ECM remodeling and suggest that ECM degradation is essential for structural synaptic plasticity.

**Keywords** ADAMTS; Perineuronal Nets; Dendritic Spines; Aggrecan; Proprotein Convertase
**Subject Category** Neuroscience

## Introduction

A challenge for the adult brain is to create new memories while maintaining stable storage of previously learned information. Memory storage relies on the persistence of neuronal circuitry, whereas learning and memory formation require functional and structural rearrangements of neuronal networks (Caroni et al, 2012). For example, it has been shown that new spines form shortly after long-term potentiation (LTP) is induced in acute hippocampal slices. During this process filopodia-like dendritic protrusions grow and become, stabilized to form new synaptic connections (Engert and Bonhoeffer, 1999; Knott et al, 2006; Maletic-Savatic et al, 1999; Yuste and Bonhoeffer, 2004). During past decades ECM has been established to form a key cellular structure that preserves the synaptic stability and inhibits synaptic structural plasticity (Dityatev et al, 2010a; Dityatev et al, 2010b). In line with this thesis the formation of ECM is inversely correlated with the decline in the structural synaptic plasticity in the developing brain (de Vivo et al, 2013; Pizzorusso et al, 2002). Moreover, acute digestion of the ECM in the visual cortex of adult animals resulted in the restoration of juvenile forms of activity-induced structural plasticity (Pizzorusso et al, 2002) and enhanced motility of dendritic spines, indicating that the perineuronal ECM may restrict the activity-induced structural plasticity of cortical dendritic spines in adult mice (de Vivo et al, 2013).

Two major forms of ECM have been described in the brain. The prominent lattice-like ECM structures that form around parvalbumin-positive (PV+) interneurons are known as perineuronal nets (PNN) (Celio and Blumcke, 1994; Lupori et al, 2023). These PNN have been shown to regulate neuronal network activity and memory formation by modulating the excitability of PV+ cells (Balmer, 2016; Dityatev et al, 2007; Favuzzi et al, 2017; Hayani et al, 2018; Rowlands et al, 2018; Wingert and Sorg, 2021). A second, ubiquitous form of the ECM is the "loose" ECM, which appears less condensed compared to the PNNs. This loose ECM is present throughout the brain, surrounding cell bodies, dendrites and synapses not only of inhibitory but also excitatory neurons (Dityatev et al, 2010b; Sorg et al, 2016; Sterin et al, 2024). It rarely fully colocalizes with synapses, but rather enwraps them leaving the synaptic cleft devoid of ECM (John et al, 2006; Tewari et al, 2024; Valenzuela et al, 2014). The perisynaptic loose ECM was suggested to contribute to a physical compartmentalization of axonal and dendritic segments. In line with this, the loose ECM around excitatory neurons in the hippocampus and cortex controls lateral

[1]Leibniz Institute for Neurobiology (LIN), 39118 Magdeburg, Germany. [2]Department of Psychiatry and Psychotherapy, University Hospital Erlangen, Friedrich-Alexander-Universität Erlangen-Nürnberg, Erlangen, Germany. [3]Department of Biology, Animal Physiology, Friedrich-Alexander-Universität Erlangen-Nürnberg, Erlangen, Germany. [4]German Center for Neurodegenerative Diseases (DZNE), Molecular Neuroplasticity Group, 39120 Magdeburg, Germany. [5]Center for Behavioral Brain Sciences (CBBS), 39106 Magdeburg, Germany. [6]Medical Faculty, Otto von Guericke University, 39120 Magdeburg, Germany. [7]Present address: Center for Cellular and Molecular Therapeutics (CCMT), Colket Translational Research Building, Children's Hospital of Philadelphia (CHOP), 3501 Civic Center Blvd, Philadelphia, PA 19104, USA. ✉E-mail: renato.frischknecht@fau.de

diffusion of AMPA receptors, surface expression of NMDA-receptor subunit 2B (GluN2B), short-term plasticity (STP) and L-type voltage Ca$^{2+}$ channel-dependent long-term plasticity (LTP; Frischknecht et al, 2009; Kochlamazashvili et al, 2010; Schweitzer et al, 2017)

Although the two forms of perineuronal ECM differ in their appearance, they share several molecular components. Both consist of a meshwork of glycoproteins and proteoglycans assembled around the long molecules of glycosaminoglycan hyaluronic acid (Fawcett et al, 2022; Gundelfinger et al, 2010). Main constituent are chondroitin sulfate proteoglycans (CSPG) of the lectican family, which are, due to the sulfation pattern of their chondroitin sulfate side chains, responsible for the non-permissive properties of the ECM for structural plasticity (Fawcett and Kwok, 2022; Foscarin et al, 2017; Gundelfinger et al, 2010; Sorg et al, 2016). Lecticans contain an N-terminal globular (G1) domain that binds to hyaluronic acid, an interaction that is stabilized by proteins of the HAPLN family (hyaluronan and proteoglycan link protein; Bekku et al, 2003; Carulli et al, 2010). Furthermore, lecticans comprise a central portion with chondroitin sulfate (CS) attachment sites, as well as a C-terminal domain (G3). This domain interacts with other ECM molecules, such as trimeric tenascin-R, and adhesion molecules on the cell surface. This bridges the core of the ECM to the cell surface (Fawcett et al, 2022; Frischknecht and Seidenbecher, 2012; Hedstrom et al, 2007). The lectican aggrecan is a core component of the PNNs, which are absent when it is knocked out in the brain (Grodem et al, 2025; Rowlands et al, 2018). The CNS specific lectican brevican is highly abundant throughout the adult brain and found in both, the loose ECM and the PNN (Deepa et al, 2006; Sorg et al, 2016). In contrast to aggrecan, loss of brevican had only little impact on PNN structure (Brakebusch et al, 2002).

Outside of the nervous system, the main mechanism controlling lectican turnover is proteolysis by specific enzymes (Bonnans et al, 2014). Brevican, aggrecan, versican and possibly also neurocan are susceptible to proteolytic cleavage by metalloproteases of the ADAMTS family (A Disintegrin and Metalloproteinase with Thrombospondin motifs) and especially by ADAMTS-4 and -5 (Lark et al, 1995; Nakamura et al, 2000; Westling et al, 2004). Proteolytic cleavage of lecticans occurs at the central unstructured region of the protein, separating the N-terminal hyaluronic acid binding G1-domain from the rest of the molecule, thereby loosening the ECM structure (Zimmermann and Dours-Zimmermann, 2008). Generation of proteolytic fragments of lecticans has been documented in brain pathology (Mayer et al, 2005; Viapiano et al, 2008). Brevican has been also reported to be proteolytically cleaved during states of homeostatic plasticity and after dopamine application (Mitlohner et al, 2020; Valenzuela et al, 2014). Interestingly, several studies revealed role of proteolytic cleavage of ECM and adhesion molecules, such as agrin and neuroligin-1, by proteases belonging to the matrix metalloproteinase (MMP) and a-disintegrin-like and metalloprotease (ADAM) families, as well as serine proteases, in the regulation of neuronal plasticity (Dityatev et al, 2010a; Matsumoto-Miyai et al, 2009; Peixoto et al, 2012; Suzuki et al, 2012).

Here, we wanted to address, whether the cleavage of ECM lectican brevican also occurs upon activity-induced neuronal plasticity and whether it could represent an endogenous mechanism to disperse the ECM allowing for structural plasticity in the adult brain to promote neuronal network rearrangements and learning. To test this hypothesis, we induced chemical LTP (cLTP) in acute hippocampal slices of young adult rats and quantified the proteolytic cleavage of the most abundant lectican brevican using semi-quantitive Western blots. We observed a marked increase in cleavage of brevican and related lectican aggrecan shortly after cLTP induction, which was abolished by ADAMTS-4/-5 protease inhibitors. Additionally, brevican cleavage at the cleavage site specific for ADAMTS-4/5 required the activity of furin proprotein convertases and depended on Ca$^{2+}$ signaling through NMDA receptors and voltage-gated calcium channels. Prevention of brevican processing reduced cLTP-induced formation of spine-like dendritic structures without affecting NMDA-dependent LTP. This suggests that neuronal activity-induced remodeling of the perisynaptic ECM was necessary for structural but not functional plasticity in the hippocampus. We propose that neuronal activity triggers a proteolytic cascade that leads to degradation of specific components of the perisynaptic ECM opening a time window for synaptic rearrangements necessary for learning and memory formation.

## Results

### cLTP promotes the proteolytic cleavage of brevican

To investigate a potential activity-dependent cleavage of the abundant brain-specific ECM proteoglycan brevican, chemical long-term potentiation (cLTP) was induced by 15 min long application of picrotoxin, forskolin, and rolipram (PFR) to acute hippocampal slices from rats aged 8–12 weeks. At this age, the mature ECM is fully established. This well-characterized treatment induces a robust potentiation of the neurotransmission in the CA3-CA1 Schaffer collateral pathway, an enrichment of GluR1 subunit of AMPA receptors at synapses and growth of dendritic spines. Previous work has also demonstrated that cLTP can be blocked by NMDA receptor antagonist APV. Thus, cLTP induces similar effects as the pairing induced LTP and shares common mechanisms (Kopec et al, 2006; Matsumoto-Miyai et al, 2009; Otmakhov et al, 2004a; Otmakhov et al, 2004b; Patterson et al, 2001). As expected, cLTP induced increased fEPSP in hippocampal CA1 evoked by stimulation of Schaffer collaterals (Fig. EV1A,B). To analyze the effect of cLTP on ECM over time, we extracted the ECM at different timepoint after cLTP induction by using chondroitinase ABC, a glycosidase that digests chondroitin sulfate, releasing the loose ECM and PNNs into the supernatant (Fig. EV2; Deepa et al, 2006; Niekisch et al, 2019). This procedure allows the extraction of extracellular proteins from cellular components by centrifugation and is necessary for separation of heavily glycosylated components of the ECM on Western blots. To evaluate the secretion and cleavage of brevican, we semi-quantitatively analyzed brevican immunoreactivity over time using an antibody directed against the N-terminus of brevican (Fig. 1B, C). This antibody detects both the full-length protein of 145 kDa and N-terminal proteolytic fragments of approximately 53 kDa (Ms α BC; Fig. 1C; black and green arrow, respectively). Abundance of the 53 kDa proteolytic fragment showed a two-fold increase immediately after cLTP stimulus, was still significantly increased 45 min later (Fig. 1D, green line) and declined gradually to basal levels within 180 min. The full-length

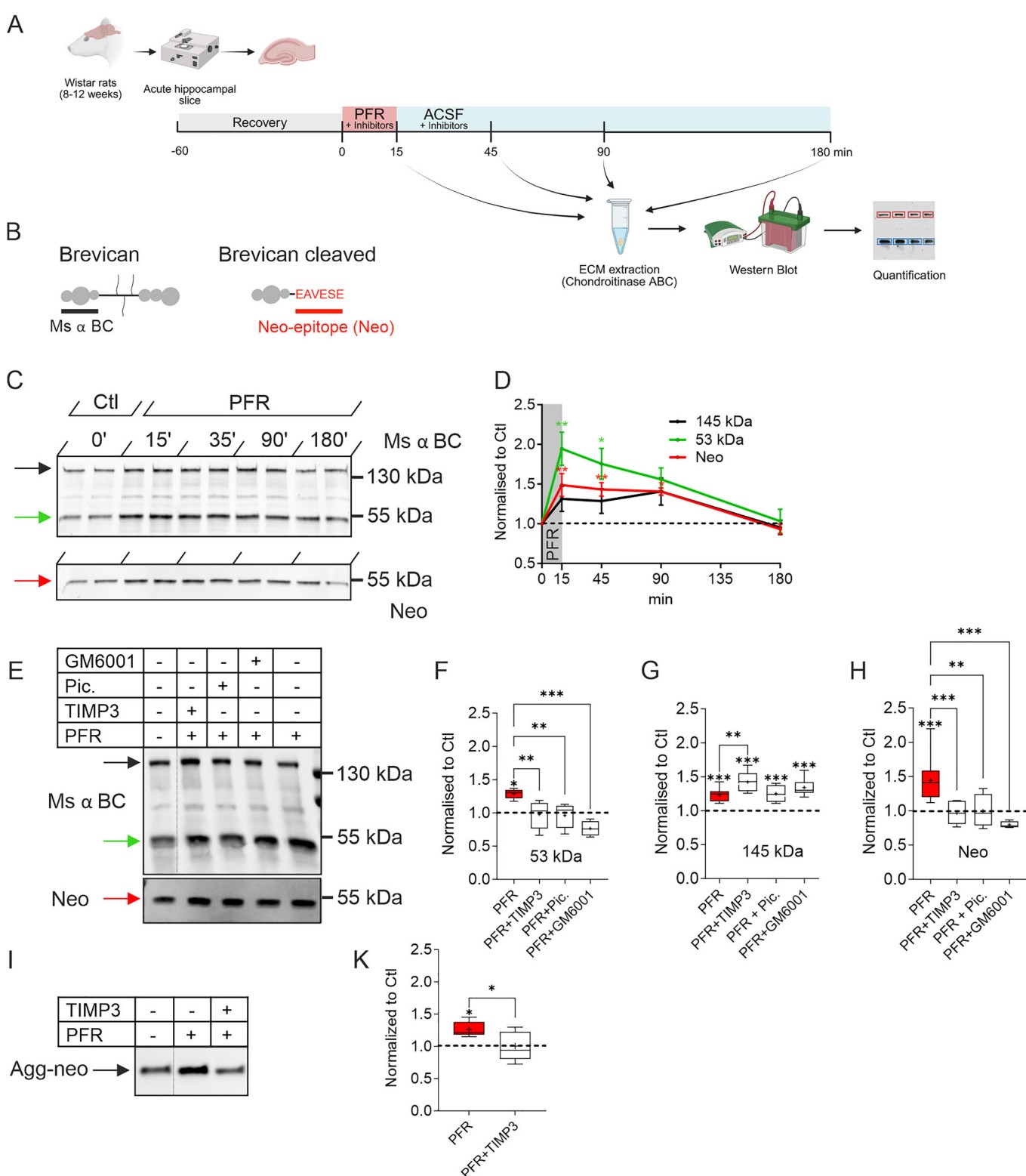

protein of 145 kDa showed a 30% increase 15 min after cLTP but did not reach statistical significance in this series of experiments (Fig. 1D, black line). Brevican undergoes proteolytic cleavage at a single site between the amino acids $Glu^{395}$-$Ser^{396}$ by members of the ADAMTS metalloprotease family (Nakamura et al, 2000).

Antibodies directed against the resulting C-terminal neo-epitope (neo) specifically recognize the 53 kDa ADAMTS-derived proteolytic fragment (Fig. 1C, red arrow; Matthews et al, 2000; Valenzuela et al, 2014). Please note that addition of a serine residue at the C-terminus of the neo epitope interferes with the binding of the neo

Figure 1.  Brevican is proteoytically cleaved in an activity-dependent manner.

(A) A schematic representation of the experimental strategy. cLTP was induced in rat acute hippocampal slices using PFR treatment for 15 min. Samples were collected at the indicated time points (Created in BioRender. https://BioRender.com/vg8nf0s). (B) Binding sites of brevican antibodies (black line: binding region of ms anti-brevican; red line: rb anti neo epitope). (C) Representative Western blots of ECM extracts from control and PFR-treated hippocampal slices probed with mouse anti-brevican antibody (top) and neo antibody (bottom). The black arrow indicates the 145 kDa full-length protein and the green and red arrows indicate the cleaved 53 kDa fragment. Quantification of the indicated bands is shown in (D). There is an immediate increase in the 53 kDa brevican proteolytic fragment after PFR treatment that persists for more than one hour (red and green lines). The full-length protein was also markedly, though not statistically significantly upregulated after PFR stimulation (black line). All values are normalized to corresponding controls, represented as dashed line (145 kDa vs. Ctl, 15 min: $P = 0.3231$, 45 min: $P = 0.3975$, 90 min: $P = 0.1558$, 180 min: $P = 0.9970$, $n = 3$; 53 kDa vs. Ctl, 15 min: $P = 0.0099$, 45 min: $P = 0.0304$, 90 min: $P = 0.0771$, 180 min: $P = 0.9999$, $n = 4$; Neo vs. Ctl, 15 min: 0.0029, 45 min: 0.0074, 90 min: 0.0116, 180 min: 0.9412, $n = 4$; one-way ANOVA, Dunnett's Multiple Comparison Test. Adjusted $P$ value, $*P < 0.05$, $**P < 0.01$, $***P < 0.001$). (E–H) Quantification of the 53 kDa and 145 kDa bands detected by ms α brevican and neo antibody 30 min after PFR treatment. Note the significant increase of the 53 kDa fragment after PFA treatment detected by ms anti-brevican (F) and neo (H) antibody. The administration of the protease inhibitors TIMP3, piceatannol (pic.) and GM6001 abolished the effect of PFR. (G) The intensity of the full-length 145 kDa protein was significantly increased after PFR treatment. This increase was significantly higher in the PFR-TIMP3 group compared to PFR alone. All blots were normalized to protein loading and control (dashed line) (F: Ctl vs. PFR: $P = 0.04$, PFR vs. PFR + TIMP3: $P = 0.006$, PFR vs. PFR +Pic.: $P = 0.009$, PFR vs. PFR + GM6001: $P < 0.001$. Ctl $n = 4$, PFR: $n = 6$, PFR+Pic.: $n = 5$, PFR + TIMP3: $n = 8$, PFR + GM6001 $n = 4$. G: Ctl vs. PFR, Ctl vs. PFR + TIMP3, Ctl vs. PFR+Pic. and Ctl vs. PFR + GM6001: $P < 0.001$, PFR vs. PFR + TIMP3: $P = 0.005$. Ctl $n = 13$, PFR: $n = 13$, PFR+Pic.: $n = 4$, PFR + TIMP3: $n = 5$, PFR + GM6001 $n = 8$. H: Ctl vs. PFR: $P = < 0.001$, PFR vs. PFR + TIMP3: $P = < 0.00$, PFR vs. PFR+Pic.: $P = 0.008$, PFR vs. PFR + GM6001: $P < 0.001$. Ctl $n = 11$, PFR: $n = 9$, PFR+Pic.: $n = 4$, PFR + TIMP3: $n = 6$, PFR + GM6001 $n = 4$. One-way ANOVA, Šídák's multiple comparisons test). (I, K) Western blot of ECM extracts using anti-aggrecan neo antibody (agg-neo). Quantification showed clear increase of the proteolytic band after PFR treatment. All blots were normalized to protein loading and control (K: Ctl vs. PFR: $P = 0.01$, PFR vs. PFR + TIMP3: $P = 0.01$, $n = 7$. One-way ANOVA, Šídák's multiple comparisons test, $*P < 0.05$, $**P < 0.01$, $***P < 0.001$. Box plot depicts the interquartile range (IQR, box), median is indicated as line, average as + and whiskers indicate minimal to maximal data point). For detailed statistics see Table EV1. Source data are available online for this figure.

antibody, confirming the specificity of neo antibody for epitope generated by ADAMTS metalloproteases (Fig. EV3). The semi-quantitative analyses revealed an increase of the 53 kDa neo fragment by approximately 50% immediately after cLTP induction, which remained significant as along as 30 min later before it declined to control levels (Fig. 1D). Thus, the cLTP induced an activity-dependent proteolysis of brevican, which was likely catalyzed by metalloproteases of the ADAMTS-family.

## Inhibition of ADAMTS prevents brevican processing

To verify the role of ADAMTS proteases in the activity-dependent proteolysis of brevican, we induced cLTP in the presence of broad-spectrum matrix metalloprotease inhibitors and ADAMTS protease-specific inhibitors. cLTP-induced formation of the 53 kDa brevican fragment recognized by either N-term specific or cleavage specific antibody was completely prevented in the presence of the general metalloprotease inhibitor GM6001 (Fig. 1E,F,H). Similarly, TIMP3, an inhibitor of ADAMTS-4 and -5, and piceatannol, an ADAMTS-4 inhibitor (Lauer-Fields et al, 2008), reduced brevican cleavage confirming the specificity of the observed effects (Fig. 1E,F,H). Interestingly, a significant increase in the full-length brevican protein was detected during cLTP induction in presence of the aforementioned protease inhibitors (Fig. 1G) indicating activity-induced increase in full-length brevican. The regulation of both the full-length and the 53 kDa fragment of brevican were unaffected by the application of the protein synthesis inhibitor anisomycin during cLTP induction (Fig. EV4). This indicates that the observed effects are more likely due to changes in brevican secretion and cleavage rather than activity-dependent regulation of brevican synthesis.

In addition to brevican, other lecticans expressed in the brain are also subject to proteolysis by ADAMTS-4 and -5 (Nakamura et al, 2000; Zimmermann and Dours-Zimmermann, 2008). One prominent target is the PNN enriched lectican aggrecan that has been found to be proteolytic cleaved by ADAMTS-4 and -5 at up to five locations (Caterson et al, 2000; Lemons et al, 2001). A specific

cleavage site between the amino acids Glu373 and Ala374, located after the N-terminal G1 domain, yields a fragment of approximately 50 kDa, which can be detected by a specific antibody (Agg-neo; Fig. 1I; Lemons et al, 2001). We used this antibody to probe ECM extracts of acute hippocampal slices treated with PFR and PFR + TIMP3, as in previous experiment (Fig. 1I). The level of the ADAMTS-derived 50 kDa fragment increased significantly, but this increase was reversed in the presence of the ADAMTS-4 and -5 protease inhibitor TIMP3 (Fig. 1K). Thus, these results suggest that brevican, as well as aggrecan, undergo proteolytic cleavage by ADAMTS proteases following cLTP induction.

## Brevican cleavage depends on the activity of proprotein convertases with furin-like specificity

Proteases from the ADAMTS family are first produced as inactive zymogens. Their activation relies on the removal of their prodomain by members of the proprotein convertase subtilisin/kexin type (PCSK) protease family, which have a substrate specificity similar to that of furin (Kelwick et al, 2015; Malfait et al, 2008). To evaluate the impact of PCSK on the proteolysis of brevican, three PCSK-specific inhibitors were administered during the induction of cLTP (Fig. 2A). All three PCSK inhibitors used in the experiments—Furin inhibitor I (Furini I), Furin inhibitor II (Furini II), and Proprotein Convertase Inhibitor (PCi)—inhibited the activity-dependent formation of the 53 kDa proteolytic fragment of brevican but had no effect on the full-length protein (Fig. 2A,B). This suggests that the cLTP-induced cleavage of brevican is dependent on proteolytic cascades involving PCSK-dependent ADAMTS activation.

The PCSK family members can activate their substrate both intracellularly in the secretory pathway and extracellularly (Seidah and Prat, 2012). To determine whether ADAMTS activation occurred intra or extracellularly, slices were incubated with 4-aminophenylmercuric acetate (APMA), a compound that activates metalloproteases by mechanism that bypasses the cleavage of their prodomain (Peixoto et al, 2012; Van Wart and Birkedal-

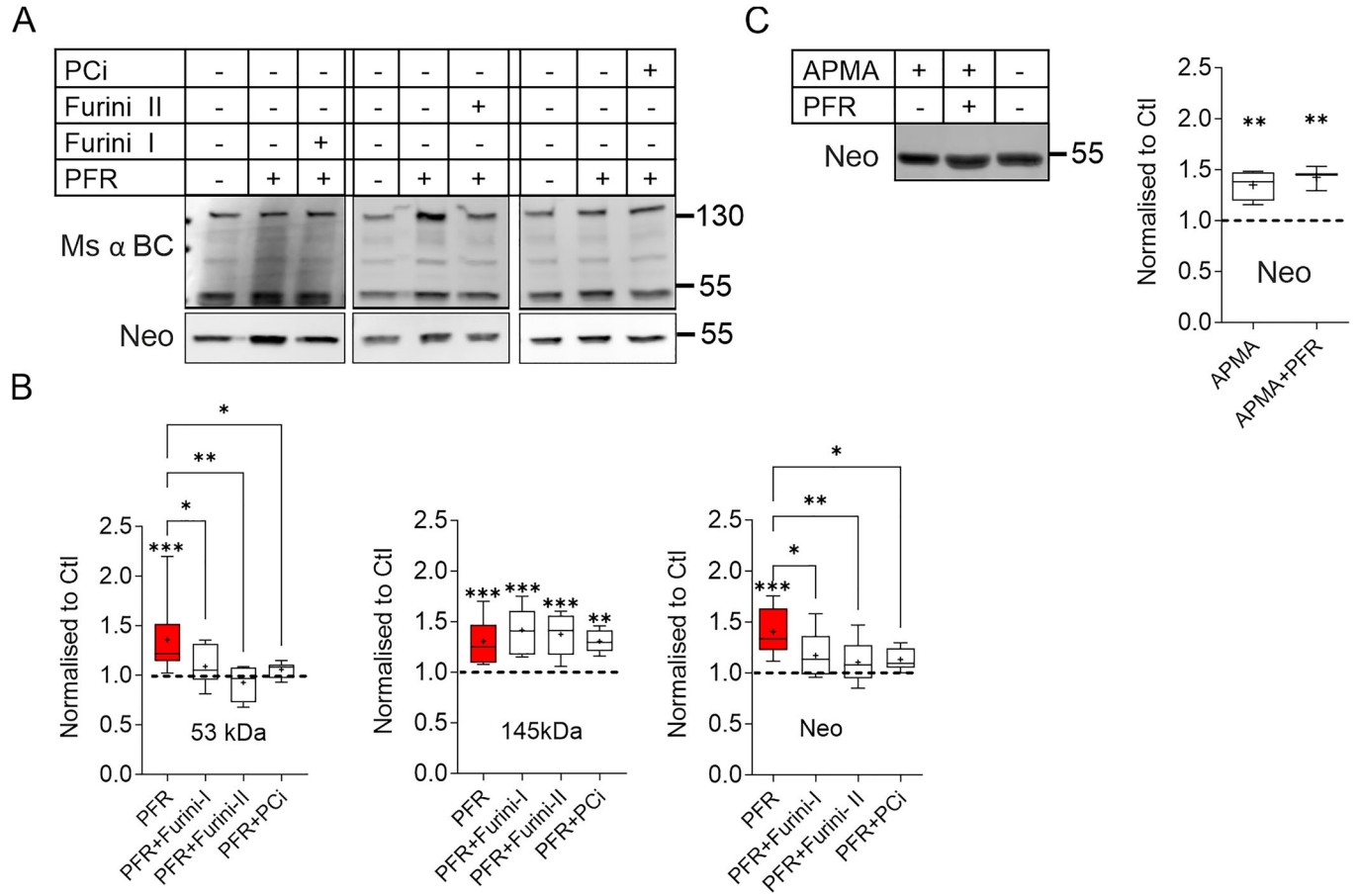

**Figure 2.  Activity of proprotein convertases is necessary to induce brevican cleavage.**

(A) Representative Western blot of extracts from acute hippocampal slices treated with PFR in combination with three different inhibitors of PCSK (Furini I, Furini II and PCi). (B) Quantification of the 53 kDa proteolytic and the 145 kDa band of brevican. All three PCSK inhibitors abolished PFR-induced brevican cleavage but had no effect on PFR-induced increase of the 145 kDa full-length protein. All data were normalized to protein loading and control slices (dashed line; 53 kDa: Ctl vs. PFR: $P = < 0.001$, PFR vs. PFR+Furini-I: $P = 0.01$, PFR vs. PFR+Furini-II: $P = 0.003$, PFR vs. PFR+PCi: $P = 0.01$. Ctl $n = 18$, PFR $n = 21$, PFR+Furini-I $n = 10$, PFR+Furini-II $n = 4$, PFR+PCi $n = 7$. 145 kDa: Ctl vs. PFR: $P < 0.001$, Ctl vs. PFR+Furini-I: $P < 0.001$, Ctl vs. PFR vs. PFR+Furini-II: $P < 0.001$, PFR vs. PFR+PCi: $P = 0.004$. Ctl $n = 16$, PFR $n = 15$, PFR+Furini-I $n = 10$, PFR+Furini-II $n = 6$, PFR+PCi $n = 6$. Neo: Ctl vs. PFR: $P < 0.001$, PFR vs. PFR+Furini-I: $P = 0.001$, PFR vs. PFR + PFR+Furini-II: $P = 0.002$, PFR vs. PFR+PCi: $P = 0.009$. Ctl $n = 15$, PFR $n = 14$, PFR+Furini-I $n = 9$, PFR+Furini-II $n = 8$, PFR+PCi $n = 7$. One-way ANOVA, Šídák's multiple comparisons test, $^*P < 0.05$, $^{**}P < 0.01$, $^{***}P < 0.001$). (C) Western blot and quantification of extracts from acute hippocampal slices treated with APMA and combined treatment with PFR. Note the significant increase of brevican cleavage under both conditions. (Neo: Ctl vs. APMA: $P = 0.0404$, Ctl vs. APMA + PFR: $P = 0.0220$. Ctl $n = 4$, APMA $n = 4$, APMA + PFR $n = 3$. One-way ANOVA, Dunnett's multiple comparisons test, $^{**}P < 0.01$. Box plot depicts the interquartile range (IQR, box), median is indicated as line, average as + and whiskers indicate minimal to maximal data point). For detailed statistics see Table EV2. Source data are available online for this figure.

Hansen, 1990). Increased brevican cleavage was evident upon addition of APMA, even without cLTP induction, as shown in Fig. 2C. The effect of AMPA was similar to the effect of cLTP and combined application of APMA and PFR resulted in effects that was not significantly different, as indicated in Fig. 2C. This result suggests that inactive ADAMTS are present in the extracellular space, and their cLTP-induced activation by proteases of PSCK family are required to initiate brevican cleavage.

## Brevican cleavage depends on D-serine

Astrocytes play a critical role in synaptic plasticity and regulate several ECM proteins, including brevican (Favuzzi et al, 2017; Hamel et al, 2005; John et al, 2006). To investigate their involvement in brevican processing during cLTP, we used

carbenoxolone (CBX), a gap junction inhibitor that has been shown to affect astrocyte coupling and disturb normal astroglial function (Beltran-Castillo et al, 2017). Important to note, CBX can also directly target neurons, affecting synaptic transmission and neuronal network activity (Rouach et al, 2003; Tovar et al, 2009). Therefore, in addition to CBX, we used an unrelated blocker of glial gap junctions, endothelin-1 (endo), which disrupts normal astrocyte function without affecting neuronal network function (Blomstrand et al, 1999; Rouach et al, 2003). The results in Fig. 3A show that cLTP-induced cleavage of brevican was completely prevented in the presence of CBX and endo indicating that normal astrocyte function is required for activity-induced brevican cleavage (Fig. 3A,B). Astrocytes support synapse function by promoting the release of D-serine, which acts as a co-agonist of NMDA receptors (NMDARs; Henneberger et al, 2010; Panatier et al, 2006; Wolosker

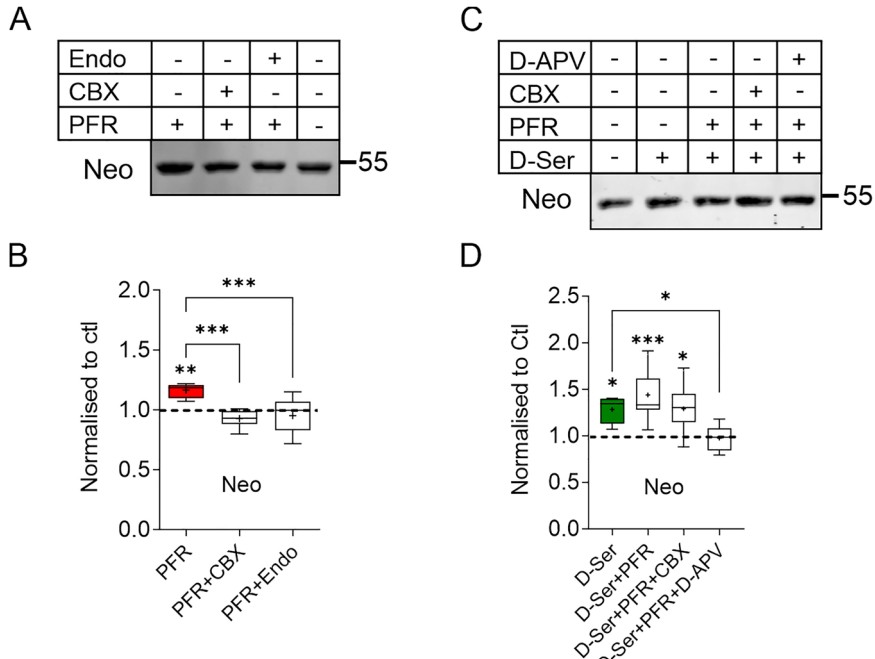

**Figure 3. Astroglia affect brevican proteolysis indirectly.**

(A) Western blot and quantification of extracts from acute hippocampal slices treated with CBX and endothelin (endo) to interfere with glia function. (B) PFR-induced augmentation of brevican cleavage was abolished by endo and CBX. All values are expressed as a percentage of the control, which is indicated by the dashed line (Ctl vs. PFR: $P = 0.0055$, PFR vs. PFR + CBX: $P = 0.0002$, PFR vs. PFR+Endo: $P = 0.0005$. Ctl $n = 8$, PFR $n = 7$, PFR + CBX $n = 6$, PFR+Endo $n = 7$. *$P < 0.05$, **$P < 0.01$, ***$P < 0.001$). (C) Western blot analysis and corresponding quantification of slices treated with D-serine, PFR, and the NMDAR blocker APV. (D) D-serine application was sufficient to induce brevican proteolysis and abolished CBX-induced block of brevican cleavage. (Ctl vs. D-Ser: $P = 0.03$, Ctl vs. D-Ser+PFR: $P < 0.001$, Ctl vs. D-Ser +PFR + CBX: $P = 0.01$, D-Ser vs. D-Ser+PFR + D-APV: $P = 0.03$. Ctl $n = 10$, D-Ser $n = 7$, D-Ser+PFR $n = 9$, D-Ser+PFR + CBX $n = 9$, D-Ser+PFR + D-APV $n = 6$. One-way ANOVA, Šídák's multiple comparisons test, *$P < 0.05$, **$P < 0.01$, ***$P < 0.001$. Box plot depicts the interquartile range (IQR, box), median is indicated as line, average as + and whiskers indicate minimal to maximal data point). For detailed statistics see Table EV3. Source data are available online for this figure.

et al, 2016). Indeed, addition of D-serine to slices completely reversed the impact of CBX on brevican cleavage. Moreover, D-serine induced brevican cleavage under basal conditions, i.e., without cLTP induction (Fig. 3B). Application of D-serine has been reported to activate postsynaptic NMDA receptors in conjunction with basal neuronal activity (Henneberger et al, 2010). To address the involvement of activation of NMDA receptors in the cleavage of brevican, we investigated the effect of pharmacological block of these receptors. The cleavage of brevican induced by D-serine was abolished by the NMDAR antagonist APV, confirming the requirement of NMDAR signaling in the process (Fig. 3C,D). These findings suggest that astrocytes are involved in the cLTP-induced cleavage of brevican. This is most likely due to their role in providing the D-serine necessary for the activation of NMDARs.

## Regulation of brevican secretion and cleavage requires activation of the postsynapse

To further analyze the role of glutamate receptors in the activity-dependent brevican cleavage, specific receptor antagonists were applied during cLTP induction. CNQX, an AMPA type glutamate receptor blocker, was found to eliminate cLTP-induced brevican processing and activity-induced increase in full-length protein (Fig. 4A–C). Treatment with the activity-dependent NMDAR

inhibitor MK801 or the blocker of GluN2B-containing NMDAR RO25-6981 inhibited cleavage but did not affect extracellular accumulation of the full-length protein (Fig. 4A–C). In addition to NMDARs, L-type voltage-gated calcium channels (L-VGCCs) also initiate $Ca^{2+}$ influx and signaling after postsynaptic depolarization in an ECM-dependent manner, which is necessary for activity-dependent neuroplasticity (Kochlamazashvili et al, 2010). To investigate the role of L-VGCCs in brevican regulation, we utilized nifedipine, an L-VGCC blocker, during the induction of cLTP. Our findings demonstrate that nifedipine diminished the extracellular upregulation of both full-length and proteolytic fragments of brevican in the slice extracts, which supports the hypothesis that postsynaptic $Ca^{2+}$ acts as second messenger in this process (Fig. 4A–C).

To further investigate this hypothesis, we utilized autocamtide-2-related inhibitory peptide (AIP), a specific inhibitor of $Ca^{2+}$/ calmodulin-dependent protein kinase II (CaMKII, Ishida et al, 1995), to block one of the primary plasticity-related $Ca^{2+}$-dependent postsynaptic signaling molecules. AIP was found to diminish the full-length form and proteolytic fragment of brevican, indicating a crucial role of $Ca^{2+}$-signaling in both the cleavage and secretion of brevican upon postsynaptic depolarization (Fig. 4D–F). While the activation of AMPAR, L-VGCC, and CaMKII regulates both the secretion and cleavage of brevican, NMDAR activation appeared to be solely responsible for its proteolysis but not for externalizing full-length brevican.

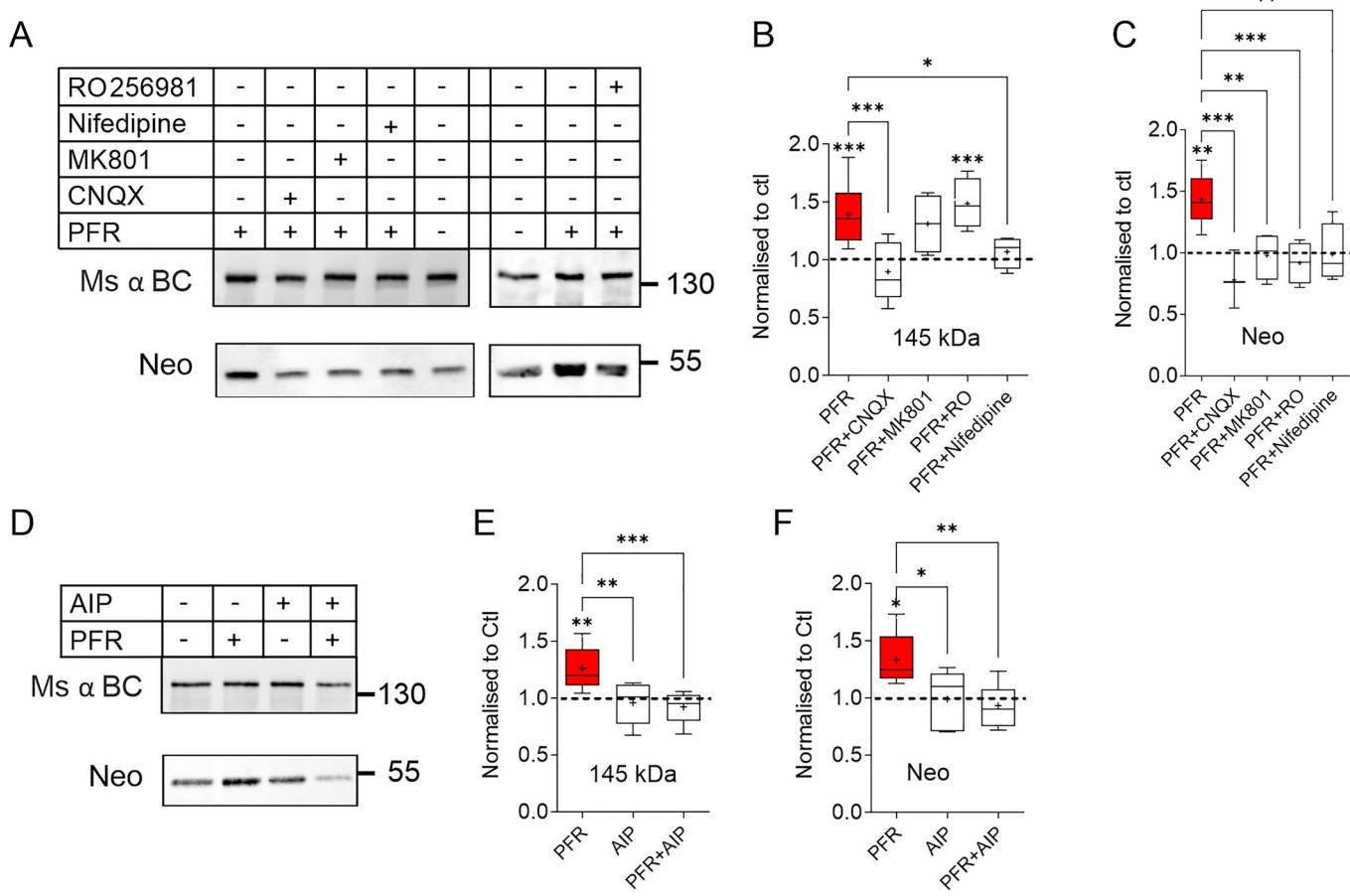

**Figure 4. Postsynaptic activation is required for brevican regulation.**

(A) Western blot and corresponding quantification of acute hippocampal slices treated with AMPA, NMDA, and L-type calcium channel blocker. (B) The PFR-induced increase of full-length brevican was not observed in the presence of CNQX or nifedipine. (C) The PFR-induced increase of the 53 kDa band is prevented in the presence of CNQX, MK801, RO256981, and nifedipine (B: Ctl vs. PFR: $P < 0.001$, Ctl vs. PFR + RO: $P < 0.001$, PFR vs. PFR + CNQX: $P < 0.001$, PFR vs. PFR+Nifedipine: $P = 0.04$. Ctl $n = 16$, PFR $n = 15$, PFR + CNQX $n = 5$, PFR + MK801 $n = 4$, PFR+Nifedipine $n = 4$, PFR + RO $n = 4$. C: Ctl vs. PFR: $P = 0.001$, PFR vs. PFR + CNQX: $P < 0.001$, PFR vs. PFR + MK801: $P = 0.003$, PFR vs. PFR + RO: $P < 0.001$, PFR vs. PFR+Nifedipine: $P = 0.004$. Ctl $n = 6$, PFR $n = 9$, PFR + CNQX $n = 3$, PFR + MK801 $n = 4$, PFR+Nifedipine $n = 4$, PFR + RO $n = 4$. One-way ANOVA, Šídák's multiple comparisons test, *$P < 0.05$, **$P < 0.01$, ***$P < 0.001$). (D) Western blot and corresponding quantification of acute hippocampal slices treated with the CAMKII blocker AIP. (E) PFR-induced increase of the full-length (145 kDa) and (F) cleaved fragment of brevican was abolished (E: Ctl vs. PFR: $P = 0.007$, PFR vs. AIP: $P = 0.006$, PFR vs. PFR + AIP: $P < 0.001$. Ctl $n = 8$, PFR $n = 7$, AIP $n = 5$, PFR + AIP $n = 8$. F: Ctl vs. PFR: $P = 0.02$, PFR vs. AIP: $P = 0.03$, PFR vs. PFR + AIP: $P = 0.004$. Ctl $n = 8$, PFR $n = 5$, AIP $n = 5$, PFR + AIP $n = 8$. One-way ANOVA, Šídák's multiple comparisons test, *$P < 0.05$, **$P < 0.01$, ***$P < 0.001$. Box plot depicts the interquartile range (IQR, box), median is indicated as line, average as + and whiskers indicate minimal to maximal data point). For detailed statistics see Table EV4. Source data are available online for this figure.

## Brevican cleavage is not necessary to induce LTP

Next, we investigated whether brevican proteolysis by ADAMTS was necessary for LTP induction. As a first approach we quantified in phosphorylation of CaMKII and extracellular signal-regulated kinase (ERK) following PFR treatment on Western blots, two molecular indicators of LTP initiation (Bayer and Schulman, 2019; Sweatt, 2001; Thomas and Huganir, 2004). We blocked ADAMTS activity during cLTP induction utilizing the protease inhibitor TIMP3. However, no changes in cLTP-induced phosphorylation of ERK and CamKII were evident (Fig. 5A–C).

Additionally, we induced early LTP in acute hippocampal slices by theta burst stimulation of the Schaffer collateral to CA3-CA1 pathway and tested the effect of ADAMTS inhibitor TIMP3 on LTP induction (Fig. 5D,E). Consistent with the biochemical results, application of TIMP3 during the LTP induction had no effect on the expression of early LTP (Fig. 5E). This suggests that ADAMTS-dependent brevican proteolysis, is not essential for LTP initiation.

## Proteolysis of brevican occurs in synaptic layers of the CA1

In the hippocampus, brevican is predominantly localized to the loose presynaptic form of ECM, while it is, to lower extent, also detectable within the PNNs surrounding parvalbumin positive interneurons (Figs. 6A,B and EV5). Aggrecan, in contrary, shows subregion-specific differences and localizes to the PNNs in the hippocampal CA1 and to the loose ECM in the dentate gyrus region of the hippocampus (Fig. 6A,B). Within CA1 brevican (detected by N-term and neo

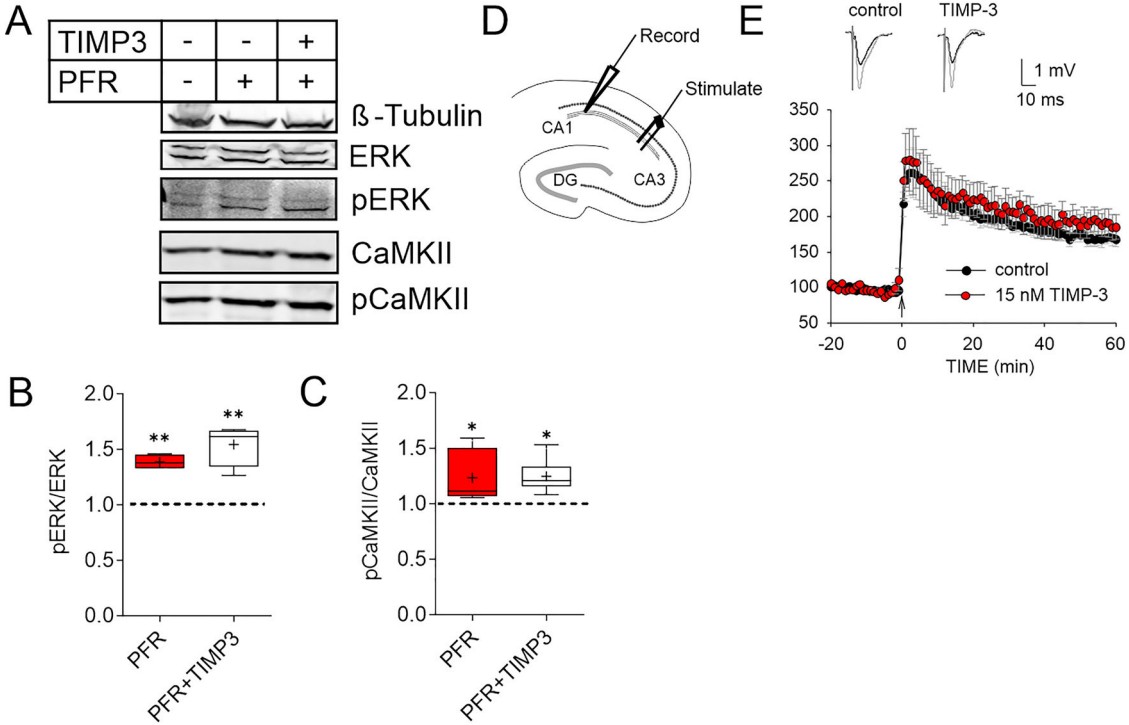

**Figure 5. Inhibition of ADAMTS proteases does not affect LTP induction.**

(A) Western blots and corresponding quantification of cell lysates from hippocampal slices treated with PFR. The ratio of phosphorylated and unphosphorylated kinases ERK and CaMKII was compared. The gel loading was adjusted according to the β-tubulin signal. (B) PFR induced ERK and (C) CaMKII phosphorylation was not affected by TIMP3 (B: Ctl vs. PFR: $P = 0.01$, Ctl vs. PFR+Timp3: $P = 0.001$. Ctl $n = 3$, PFR $n = 4$, PFR + TIMP3 $n = 4$. C: Ctl vs. PFR: $P = 0.0492$, Ctl vs. PFR+Timp3: $P = 0.0263$. Ctl $n = 6$, PFR $n = 7$, PFR + TIMP3 $n = 7$. One-way ANOVA, Šídák's multiple comparisons test, *$P < 0.05$, **$P < 0.01$, ***$P < 0.001$. Box plot depicts the interquartile range (IQR, box), median is indicated as line, average as + and whiskers indicate minimal to maximal data point). (D) Recordings of field excitatory postsynaptic potentials (fEPSPs) in the *stratum radiatum* of the CA1b were obtained. Stimulation pulses were applied to Schaffer collaterals, and LTP was induced by applying five theta bursts. (E) A robust increase in the slope of fEPSPs was observed in control (black, $n = 10$) and in slices treated with TIMP3 (red, $n = 7$). For detailed statistics see Table EV5. Source data are available online for this figure.

antibodies) localizes to the vicinity of the synaptic endings marked by Bassoon antibody (Fig. 6C–E), which is in line with previous publications (Mitlohner et al, 2020; Valenzuela et al, 2014). Previous experiments using Western blotting of ECM extracts from hippocampal slices showed that brevican was proteolytically cleaved in an activity-dependent manner. However, these experiments provide no spatial information about where brevican was processed. Therefore, we quantified immunoreactivity of neo antibody to identify the ADAMTS-derived N-terminal fragment of brevican within the *stratum radiatum* of control and PFR-treated acute hippocampal slices (Fig. 6F,G; boxed area). We observed elevated fluorescence intensities after PFR treatment, indicating cleavage of brevican within the loose ECM present in the molecular layer of the CA1 (Fig. 6H). Next, we wondered whether brevican was processed in the vicinity of excitatory synapses. To that end, we stained acute hippocampal control and PFR-treated slices with Homer1, which is a component of the postsynaptic density of excitatory synapses, as well as with brevican neo-epitope antibody (Fig. 6I). Quantification of neo-epitope florescence intensity within synaptic regions marked by Homer1 immunoreactivity, showed a significant increase in PFR treated slices (Fig. 6K). Together, these experiments indicate an activity-dependent presynaptic proteolysis of brevican by members of the ADAMTS family in hippocampal CA1 *stratum radiatum*.

## Formation of cLTP-induced dendritic protrusions requires the activity of ADAMTS proteases

LTP induces formation of new synapses, which is preceded by the formation of dynamic filopodia-like dendritic protrusions, of which some are stabilized and from new synaptic connections (Engert and Bonhoeffer, 1999; Jourdain et al, 2003). To visualize LTP-induced structural plasticity, we took advantage of a mouse strain expressing YFP in sparse neurons within the CA1 pyramidal layer. In this strain, YFP-positive excitatory pyramidal neurons within CA1 can be identified morphologically as cells with pyramidally shaped cell body and long apical dendrites with small collateral branches within the *stratum radiatum*. Their fine structural features like dendritic spines can be easily visualized (Fig. 7A,B). These neurons and their dendrites were embedded in loose brevican-containing ECM and were devoid of typical PNN (Fig. 7B). The expression of YPF in these cells allows reliable measurement of the length and number of dendritic protrusions on secondary dendrites within the *stratum radiatum* (Fig. 7C). Thirty minutes after the induction of cLTP, a significant increase in dendritic protrusions number and length was evident in the PFR-treated slices compared to the controls (Fig. 7D–G), which is consistent with previous report (Matsumoto-Miyai et al, 2009). However, the

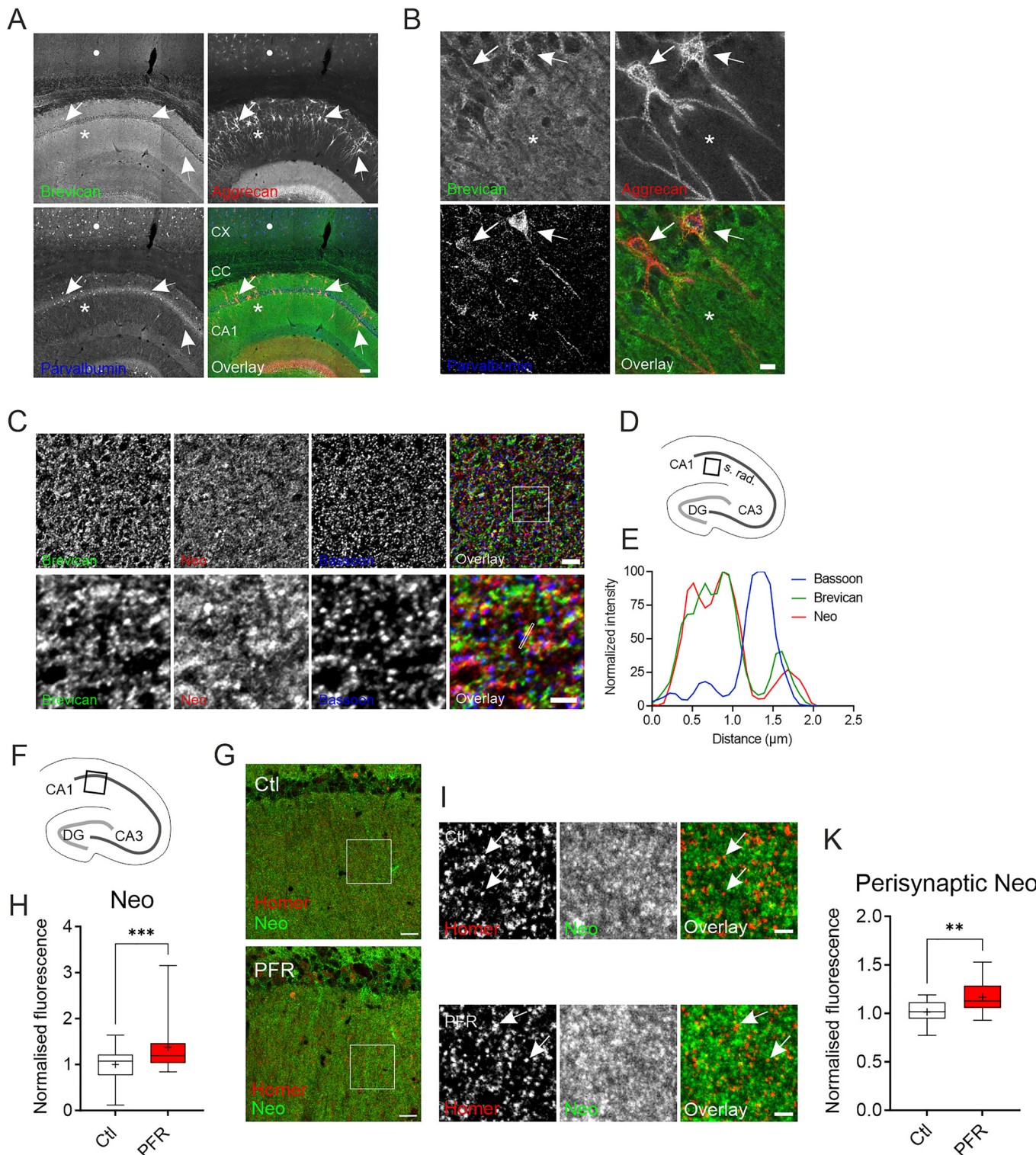

same treatment in the presence of the protease inhibitor TIMP3 failed to induce any change in the number and size of dendritic protrusions were identical to those in control slices (Fig. 7D–G). This indicates that the activity of proteases from the ADAMTS family was necessary for cLTP-induced structural plasticity of hippocampal dendritic protrusions.

## Discussion

The aim of this study was to investigate the activity-dependent proteolysis of brevican, a prototypical component of the "loose" perineuronal ECM, the molecular and cellular players involved in this process, and its relevance for functional and structural activity-

◀

**Figure 6. Brevican is cleaved within stratum radiatum and perisynaptically.**

(A) Staining of coronal sections of paraformaldehyde-fixed adult rat brains using antibodies against brevican, aggrecan, and parvalbumin (blue). Brevican (green) is distributed diffusely in the cortex (white dot) and in the synaptic layers of the hippocampus (asterisks). Brevican staining is also present in perineuronal nets labeled with aggrecan (red) and positive for parvalbumin (blue, arrows in **A** and **B**). CX: cortex; CC: corpus callosum; CA1: cornu ammonis area 1; DG: dentate gyrus. Note the strong, diffuse labeling of brevican in the stratum radiatum (asterisk in **A** and **B**; scale bar: 100 µm). (B) Higher magnification of A (scale bar: 10 µm). (C) Cryosection of adult rat hippocampus stained with anti-brevican (green) brevican neo (neo, red) and Bassoon (blue) antibody. Top: stratum radiatum (s. rad) indicated in the scheme in (**D**) (scalebar 5 µm). Boxed area in the overlay is shown in higher magnification below (scalebar 2 µm). (E) Fluorescence intensity profile of Bassoon, brevican and neo as indicated in the overlay image in C. Note the perisynaptic staining of brevican and brevican neo. (F–K) Brevican neo (green) and Homer1 (red) immunofluorescence in acute hippocampal slices in ctl and after PFR treatment (**G**). The total fluorescence intensity in the s. rad of the CA1 region was compared (boxed area; scale bar, 25 µm). (H) A significant increase of neo staining was observed following PFR treatment within the full region (Ctl vs. PFR: $P < 0.0001$. Ctl $n = 71$, PFR $n = 73$. Unpaired t test, ***$P < 0.001$). (I) High magnification image of the acute hippocampal slices depicted in (**K**). Arrows indicate individual synaptic puncta. (K) Quantification of neo epitope fluorescence intensity at Homer1 puncta (Ctl vs. PFR: $P = 0.005$. $n = 16$. Unpaired t test, **$P < 0.01$. Box plot depicts the interquartile range (IQR, box), median is indicated as line, average as + and whiskers indicate minimal to maximal data point). For detailed statistics see Table EV6. Source data are available online for this figure.

induced synaptic plasticity. We demonstrated the cleavage of perisynaptic brevican during cLTP induction in the synaptic layers of CA1 region in acute hippocampal slices of adult rats. The processing of brevican is conducted by proteases belonging to the ADAMTS family, which are activated in the extracellular space by PSCKs. Initiation of this proteolytic cascade depends on NMDA receptors, L-type VDCCs, and CaMKII. Interestingly, inhibiting proteases of the ADAMTS family did not disrupt early LTP. However, it reduced the number of dendritic protrusions typically formed after cLTP induction. Thus, perineuronal ECM modulation appears to occur in an activity-dependent manner and is a prerequisite for structural, but not functional plasticity.

## cLTP activates proteolytic cascade to cleave ECM lectican brevican

Activity-dependent extracellular proteolysis and the resulting changes in the ECM have the potential to enhance neuronal plasticity by altering cell adhesion and ECM architecture (Dityatev et al, 2010a). Indeed, previous studies have demonstrated that ECM proteins and cell adhesion molecules, including agrin, dystroglycan, neuroligin-1 and CD44, undergo proteolytic cleavage in an activity-dependent manner that affects neuronal network architecture and learning (Bijata et al, 2017; Ferrer-Ferrer et al, 2023; Figiel et al, 2022; Matsumoto-Miyai et al, 2009; Peixoto et al, 2012). The present study demonstrates that cLTP induces processing of the ECM lectican brevican at perisynaptic sites, indicating that the perineuronal ECM undergoes rapid, activity-dependent remodeling. Our data indicated a key role of ADAMTS proteases in this process. We observed (1) cLTP-induced increase in the brevican and aggrecan fragments with of the ADAMTS-4 and -5 specific cleavage sites and (2) addition of three independent inhibitors of protases of ADAMTS family completely blocked the cLTP-induced brevican and/or aggrecan cleavage. We have recently shown, that ADAMTS-dependent brevican proteolysis also occurred after prolonged neuronal network silencing that promote homeostatic synaptic scaling, or by activation of D1-like dopamine receptors (Mitlohner et al, 2020; Valenzuela et al, 2014).

For local and regulated proteolysis of ECM, activity of the specific proteases must be tightly controlled. This can be achieved by (1) regulating their synthesis, (2) their cLTP-induced externalization from internal stores or (3) regulating their activation. Application of anisomycin, a potent blocker of protein synthesis, during cLTP induction did not alter brevican cleavage, indicating

that de novo translation is not required for cLTP-induced brevican proteolysis (see Fig. EV4). cLTP could induce activation of ADAMTS within the secretory pathway prior its secretion. However, an incubation of acute hippocampal slices with the compound APMA that activates matrix metalloproteases non-proteolytically (Peixoto et al, 2012; Van Wart and Birkedal-Hansen, 1990) led to increase in brevican cleavage, which speaks against this alternative. Interestingly, the members ADAMTS family have been reported to bind sulfated glycosaminonglycans on lecticans like aggrecan or brevican (Flannery et al, 2002). Thus, we propose that the ADAMTS proteases are located in their inactive form at their place of action, and can, rapidly cleave ECM lecticans upon their activation to create space for structural rearrangements.

To prevent their ectopic activity proteases of the ADAMTS family are known to be synthetized as inactive zymogens. To activate their enzymatic activity, their pro-domain must be removed, for example by pro-protein convertases (PCSK) with furin-like specificity that recognize a specific sequence within the pro-domains of ADAMTS (Kelwick et al, 2015). Nine PCSKs have been identified of which seven comprise a furin-like specificity, making them good candidates for ADAMTS activation (Seidah and Prat, 2012). Indeed, PCSKs were suggested to activate ADAMTS in cultured neurons previously (Lemarchant et al, 2014; Mitlöhner et al, 2020; Tortorella et al, 2005). We have used three independent inhibitors of PCSKs with furin-like activity and all three prevented cLTP-induced brevican cleavage further supporting the idea that activation of preexisting enzymes leads to local remodeling of perisynaptic ECM. The activation of brevican cleavage by proteolytic cascade could lead to exponential amplification of the proteolytic activity allowing rapid and efficient ECM remodeling.

Of note, our results revealed an essential contribution of astrocytes in the brevican proteolysis. However, they do not play a direct role in brevican cleavage, such as through secretion of ADAMTS proteases, but they are rather required for proper activation of NMDA receptors as experiment with D-serine supplementation have revealed.

## ECM remodeling is necessary for structural plasticity but does not affect early LTP

Dendritic spines are highly dynamic structures representing the postsynaptic site of most excitatory synapses. They can change in length or thickness and be eliminated or formed anew in an activity-dependent manner (Sala and Segal, 2014). Spine dynamics

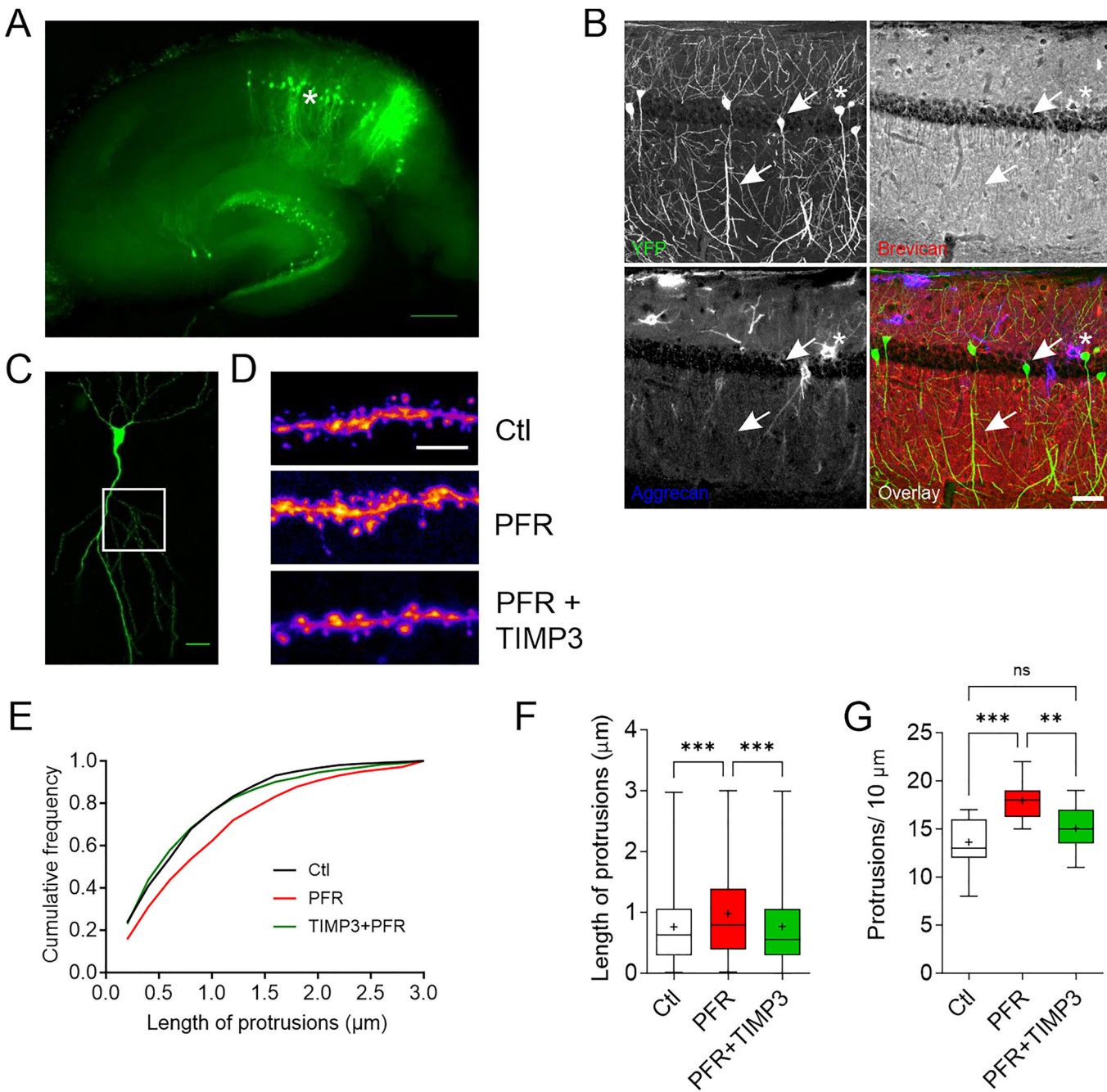

**Figure 7. The formation of dendritic protrusions in response to PFR is inhibited by TIMP3.**

(A) Acute hippocampal slice from a mouse expressing YFP in sparse neurons. Asterisk indicates CA1 region (scale bar 500 μm). (B) YFP-positive neurons and their neurites (arrows) are surrounded by loose brevican-containing ECM and lack strong labeling with the PNN marker protein aggrecan (asterisks). (C) Example of YFP expression in CA1 neuron. Dendritic spine-like structures were counted from the secondary dendrite (Boxed area, scale bar 20 μm). (D) Examples of dendrites used for quantification. Note the long spine-like structures in PFR-treated slices (arrow, scale bar 5 μm). (E) Cumulative distribution of spine-like dendritic protrusion length from control (black), PFR (red) and PFR + TIMP3 (green) treated slices. (F) Quantification of the length of spine-like protrusions (Ctl vs. PFR: $P < 0.001$, Ctl vs. PFR + TIMP3: $P < 0.001$. Ctl $n = 462$, PFR $n = 1140$, PFR + TIMP3 $n = 675$. One-way ANOVA, Šídák's multiple comparisons test, $*P < 0.05$, $**P < 0.01$, $***P < 0.001$). (G) Quantification of number of protrusions/10 μm dendrite (Ctl vs. PFR: $P < 0.001$, Ctl vs. PFR + TIMP3: $P = 0.004$. Ctl $n = 19$, PFR $n = 16$, PFR + TIMP3 $n = 17$. One-way ANOVA, Šídák's multiple comparisons test, $*P < 0.05$, $**P < 0.01$, $***P < 0.001$. Box plot depicts the interquartile range (IQR, box), median is indicated as line, average as + and whiskers indicate minimal to maximal data point). For detailed statistics see Table EV7. Source data are available online for this figure.

are crucial for forming, refining, and readjusting functional synaptic networks in the process of structural synaptic plasticity, which decisively impacts information processing in the brain. This process is particularly pronounced in the brains of young animals, as developing neuronal circuits are refined in an activity-dependent manner. Although the adult brain is considered less plastic than younger brains, the rearrangement of dendritic spines has been observed and linked to circuit remodeling (Berry and Nedivi, 2017; Zuo et al, 2005). Interestingly, the transition from the periods of high plasticity in young animals to adulthood coincides with the formation chondroitin sulfate containing ECM indicating that ECM might play a role in the structural stabilization of the circuitry during development (Hou et al, 2017; Pizzorusso et al, 2002). In line with this thesis, reinstatement of juvenile forms of plasticity and increased motility of dendritic spines within the visual cortex was observed upon enzymatic removal of ECM (de Vivo et al, 2013). Injections of the serine protease tissue type plasmin activator (tPA) that among other also leads to digestion of ECM proteins has similar effects (Mataga et al, 2004; Oray et al, 2004). Endogenous ECM weakening may be a mechanism to increase transiently structural plasticity in the adult brain to maintain the capacity to modify existing neuronal networks and allow for learning and memory formation.

Here we provide evidence that brevican cleavage by ADAMTS proteases is necessary to induce dendritic protrusions after cLTP induction, underlining the importance of extracellular proteolysis of ECM for structural plasticity. Number of studies suggested an importance of extracellular proteolysis for activity-dependent structural plasticity (Dityatev et al, 2010a). For instance, the matrix protease MMP-9, which is secreted in an activity-dependent manner and can cleave the cellular ECM receptor CD44, has been implicated in the structural and functional plasticity upon induction of LTP (Bijata et al, 2017; Magnowska et al, 2016; Michaluk et al, 2009; Szepesi et al, 2014). Similarly, the neuronal activity-controlled secretion of serine protease neurotrypsin is required for structural plasticity upon induction of cLTP (Frischknecht et al, 2008; Matsumoto-Miyai et al, 2009; Reif et al, 2007; Stephan et al, 2008). Interestingly, the neurotrypsin-derived proteolytic fragment of agrin is required and sufficient for the effect of neurotrypsin on the cLTP-induced formation of dendritic protrusions (Matsumoto-Miyai et al, 2009; Stephan et al, 2008). Thus, ECM proteolysis does not only create space for rearrangements but also generates new signaling moieties. ADAMTS-dependent cleavage of brevican releases a 55 kDa N-terminal proteolytic fragment of brevican. It is notable that this fragment, but not the full-length protein, has been demonstrated to promote glioma cell motility and invasiveness, suggesting that this fragment may have motility-promoting effects (Hu et al, 2008; Viapiano et al, 2008). It will be interesting to investigate the role of the brevican-derived proteolytic fragments in this context.

Although necessary for activity-induced formation of dendritic protrusions, blockade of ADAMTS proteolytic activity did not affect the induction and expression of early LTP measured as long as 60 min upon its induction. A similar observation was made in neurotrypsin knockout animals, which lack activity-dependent agrin proteolysis and show normal LTP induction (Matsumoto-Miyai et al, 2009). However, while neither the broad spectrum MMP inhibitor GM6001 nor a more specific MMP-9 inhibitor affected induction of hippocampal CA1 LTP they prevented early

structural plasticity and expression of late LTP (Nagy et al, 2006). The connection of late-LTP with changes in spine morphology and formation of new synapses is well documented (Muller et al, 2002). Whether ADAMTS-dependent extracellular proteolysis of ECM components including brevican and aggrecan affect not only structural plasticity, but also late LTP remains to be experimentally addressed.

## Activity-dependent proteolysis of ECM as possible mechanism for ECM restructuring

In addition to the proteolytic cleavage of brevican, we observed an increase in full-length brevican after cLTP induction (Fig. 1D,G). This effect is slightly more pronounced, when ADAMTS or PCSK inhibitors, which prevent brevican processing, are included in the experiment (Figs. 1G and 2A,B). Replenishment of the ECM was also observed in vivo after enzymatic digestion of the ECM, highlighting a possible dynamic ECM nature during adult brain function (Happel et al, 2014). In addition, while initial training of mice in an auditory cortex-dependent learning task resulted in a reduction in brevican, successful learning correlated with marked upregulation of its expression, restoring its basal levels (Niekisch et al, 2019). Based on these data, one could speculate that proteolysis induced by neuronal activity leads to the removal of the ECM, which opens a window of opportunity for the structural and functional rearrangements necessary for neuronal plasticity. The simultaneous secretion of new ECM components enables the rapid formation of a new matrix, which is essential for consolidating activity-induced rearrangements. This hypothesis is in line with defects in neuroplasticity observed in mice where ECM was disturbed genetically or enzymatically. Deletion of brevican or neurocan affected hippocampal CA1 LTP maintenance, while tenascin R deletion or enzymatic removal of ECM by means of chondroinase ABC treatment impaired LTP induction and cognition (Brakebusch et al, 2002; Bukalo et al, 2001, 2007; Montag-Sallaz and Montag, 2003; Zhou et al, 2001). Mice lacking four components of the brains ECM brevican, neurocan, tenascin R and tenascin C showed also abnormal synaptic potentiation and depression and learning (Jansen et al, 2017; Mueller-Buehl et al, 2025).

A recent work, indicated that not only structural rearrangements of perisynaptic ECM, but also changes in their posttranslational modifications could contribute to synapse function and memory (Chelini et al, 2024). A particular sulfation pattern of the chondroitin sulfate side chains of versican emerge after sensory stimulation and coincidence with dendritic spine plasticity and depletion of versican impaired CA1 hippocampal LTP and memory in mice (Chelini et al, 2024). In another work, it has been demonstrated that components of the ECM can be removed by endocytosis and subsequently recycled in an activity-dependent manner (Dankovich et al, 2021). Recently, also microglia have been found to contribute to the ECM remodeling by engulfing ECM proteins (Nguyen et al, 2020; Strackeljan et al, 2021; Venturino et al, 2021). Depletion of microglia increased the expression of perineuronal ECM leading to changes in synapse density (Nguyen et al, 2020; Strackeljan et al, 2021).

Together these works show that multiple mechanisms can induce ECM modification and thereby affect neuronal plasticity. Extracellular proteolysis, sulfation, ECM production and recycling

may act synergistically in vivo to rapidly remove local ECM proteins. This possibly creates a window of opportunity with increased brain plasticity permissive for neuronal network readjustment, which is necessary for learning and memory formation. Subsequent replenishment of ECM may help to stabilize newly formed synapses and memories.

# Methods

### Reagents and tools table

| Reagent/resource | Reference or source | Identifier or catalog number |
| --- | --- | --- |
| **Experimental models** | | |
| Slick-V cre | (Young et al, 2008) https://www.jax.org/strain/007610 | B6;SJL-Tg(Thy1-cre/ERT2,-EYFP)VGfng |
| Sprague Dawley rats (CD IGS) | Charles River | |
| **Chemicals, enzymes and other reagents** | | |
| Autocamptide-2 | Sigma-Aldrich | SCP0001 |
| Anisomycin | Sigma-Aldrich | A9789 |
| Carbenoxolone disodium salt (CBX) | Sigma-Aldrich | C4790 |
| Chondroitinase (ChABC) | Sigma-Aldrich | C2905 |
| D-(-)-2-Amino-5-phosphonopentanoic acid (APV) | Tocris Bioscience | 0106 |
| D-serine | Merck Millipore | S4250 |
| Complete protease inhibitor | Roche | 11697498001 |
| Fluoromount-G™ | Thermo Fisher | 00-4958-02 |
| Furin Inhibitor I | Merck Millipore | 344930 |
| Furin Inhibitor II (Hexa-D-arginine) | Tocris Bioscience | 4711 |
| Endothelin-1 (endo) | Sigma-Aldrich | E7764 |
| GM6001 | Tocris Bioscience | 2983 |
| PhosSTOP phosphatase inhibitor | Roche | 04 906 845 001 |
| Picrotoxin | Sigma-Aldrich | P1675 |
| polyvinylidene difluoride (PVDF) membranes | Millipore | IPFL00005 |
| Nifedipine | Tocris Bioscience | 1075 |
| 6-Cyano-7-nitroquinoxaline-2,3-dione disodium (CNQX) | Tocris Bioscience | 1045/1 |
| Forskolin | Tocris Bioscience | 1099/10 |
| Rolipram | Tocris Bioscience | 0905/10 |
| Ro 25-6981 | Tocris Bioscience | 1594/10 |
| TIMP3 | R&D System | 973-TM |
| Tissue-Tek | Sakura | 4583 |
| 2,2,2-trichloroethanol | Merck | T54801 |

| Reagent/resource | Reference or source | Identifier or catalog number |
| --- | --- | --- |
| **Antibodies** | | |
| Aggrecan (rb) | Millipore | AB1031 |
| Aggrecan Neo (rb) | ThermoFisher | PA1-1746 |
| Bassoon (ms) | Synaptic Systems | 141111 (mab7f) |
| Brevican (ms) | BD Transduction Laboratories | 610894 |
| Brevican (gp) | John et al, 2006 | Home made |
| Brevican Neo (rb) | Valenzuela et al, 2014 | Home made |
| CaMKII (ms) | SantaCruz | sc-13141 |
| p-CaMKII (ms) | SantaCruz | sc-32289 |
| ERK (ms) | BD Biosciences | 610124 |
| p-ERK (ms) | Sigma-Aldrich | M9692 |
| Homer-1 (gp) | Synaptic Systems | 160318 |
| Parvalbumine (ms) | Sigma | P3088 |
| Synaptotagmin1 (ms) | Synaptic Systems | 105011 |
| β-tubulin (ms) | ThermoFisher | MA5-16308 |
| mouse IgG IRDye-800CW | LICORbio | 926-32210 |
| rabbit IgG IRDye-680RD | LICORbio | NC0252290 |
| rabbit 488 Alexa | Invitrogen | A32731 |
| mouse 555 Alexa Plus | Invitrogen | A32727 |
| guinea pig 647 Alexa | Invitrogen | A21450 |
| **Software** | | |
| Prism 10.0.1 | GraphPad Software, Inc. | https://www.graphpad.com/ |
| IntraCell software | LIN Magdeburg, Germany | In house development |
| Fiji (ImageJ) | | https://imagej.net/software/fiji/downloads |
| CorelDRAW 2024 | | |
| LAS AF software | Leica | Version 2.0.2 |
| Microsoft Excel | Microsoft Corporation | Version 2021 |
| SigmaPlot | Systat Software Inc. | |

### Acute hippocampal slice preparation

Experiments were carried out following the European Council Directive (2010/63/EU, amendment 2019), fully in accordance with local regulations, registered under approved und FAUTS13/2016 and by FAU TS 13/2016 and TS 2/2023. 8–12-week-old Wistar rats were anesthetized with isoflurane (4%). After decapitation, the brain was rapidly removed and immersed in an oxygenated ice-cold artificial cerebrospinal fluid (ACSF, 125 mM NaCl, 2.5 mM KCl, 1.25 mM $NaH_2PO_4$, 25 mM NaHCO3, 2 mM $CaCl_2$, 1 mM $MgCl_2$, 25 mM glucose). Hippocampi were isolated in an oxygenated ice-cold ACSF and transverse hippocampal slices (350 μm) were

prepared using a McIlwain tissue chopper (Mickle Laboratory). The slices were allowed to recover at room temperature for 90 min and then recovered at 32 °C for 1 h. Throughout the procedure, the slices were perfused with ACSF solution which was bubbled with 95% $O_2$ and 5% $CO_2$.

## Induction of cLTP in acute hippocampal slices

Chemical LTP (cLTP) was induced with a combination of picrotoxin (50 μM), forskolin (50 μM) and rolipram (0.1 μM; PFR) in ACSF without $Mg^{2+}$ for 15 min at 32 °C as previously described (Matsumoto-Miyai et al, 2009). All the inhibitors were incubated 20–30 min before stimulation and were present until the end of the recovery after the chemical stimulation (Figs. 1A and 2A).

## Sample preparation and semiquantitative Western blot

For ECM extraction acute hippocampal slices were incubated for 30 min at 37 °C in extraction buffer containing chondroitinase ABC (0.2 U/ml) in 0.1 M Tris-HCl, pH 8.0, 0.03 M sodium acetate, complete protease inhibitor cocktail and PhosSTOP phosphatase inhibitor cocktail. For three slices 90 μl of the ECM extraction buffer was used. Slices were triturated and centrifuged at $14,000 \times g$ for 10 min. The supernatant was collected and mixed with 4 x SDS Laemmli buffer and stored at −80 °C. Protein concentration was determined by the Amido Black colorimetric assay. For SDS-PAGE, gradient gels (tris-glycin 5–20%) containing 2,2,2-trichloroethanol (TCE) were used to visualize and quantify the total amount of protein (Ladner et al, 2004). Each fraction was loaded twice and gels were blotted onto polyvinylidene difluoride membranes using the Mighty Small Transfer Tank system (Hoefer). PVDF membranes were blocked in TBS-T containing 0.1% Tween20 and 5% bovine serum albumin for 30 min at room temperature. Primary antibodies were diluted in TBS-T containing 2% BSA and probed for 2 h at room temperature or overnight at 4 °C. Membranes were subsequently incubated with fluorescence coupled secondary antibodies. Blots were scanned using the LI-COR Odyssey system and signals were quantified using ImageJ. Signal intensity was corrected for protein load as determined by TCE for ECM extracts, with GAPDH or β-actin for cell lysates before normalization to the average value of the control group loaded on each membrane.

## Immunocytochemistry

The acute hippocampal slices were fixed overnight in PBS containing 4% PFA and 4% sucrose. They were then placed in 30% sucrose for cryoprotection and frozen in Tissue-Tek embedding medium. The slices were then resliced using a cryostat (Leica CM 3050 S) into 30-μm sections, permeabilized in phosphate-buffered saline (PBS) containing 0.3% Triton X-100 and 10% fetal calf serum (FCS) for one hour at room temperature. Antibodies were incubated for 72 h at 4 °C, and the secondary antibodies were incubated overnight at 4 °C. Between each staining step, the slices were washed three times for ten minutes in phosphate-buffered saline (PBS). The slices were then mounted on a glass slide. Images were captured using a Leica SP5 confocal microscope (63x objective, NA 1.40, or 20x objective, NA 0.8). For quantitative immunofluorescence, 10 to 12 optical sections were taken with a

step size of 250 nm. Maximum intensity projections were generated using the Fiji (ImageJ) software. Fluorescence intensity was quantified within 50 μm rectangular regions within the molecular layer of the CA1 region using the ImageJ software. Coronal sections (14 μm) were taken from PFA-fixed adult rat brains using a cryostat, blocked and stained free-floating in the aforementioned buffers.

## Quantification of dendritic protrusions

For the quantification of dendritic protrusions, acute hippocampal slices from adult Slick V-cre YFP-expressing mice were utilized, as these mice express YFP in sparse neurons. Slices were prepared for imaging as previously described. For the analysis of dendritic protrusions, images of apical secondary dendrites 80 μm from the soma were obtained using a Leica SP5 confocal microscope (63x objective, NA 1.40). The images were processed and protrusions were quantified using Fiji software.

## Statistical analysis

Statistical analyses and graphical representations were generated using GraphPad Prism 10. For statistical comparison between two groups unpaired-sample t tests were used. Statistical comparison of multiple groups was performed by an analysis of variance (one-way ANOVA) followed by Šídák's multiple comparisons test for post-hoc comparisons as indicated in figure legends. Collected numerical data are presented in expanded view tables (Table EV1). Data is presented as box with whiskers (25th to 75th percentiles), whiskers represent min to max, the line indicates median and mean is shown as '+'.

## LTP recordings in hippocampal slices

Each mouse was euthanized by cervical dislocation, followed by decapitation. The brain was extracted from the skull and transferred into ice-cold ACSF, saturated with carbogen (95% $O_2$/5% $CO_2$) containing (in mM) 250 sucrose, 25.6 NaHCO₃, 10 glucose, 4.9 KCl, 1.2 KH₂PO₄, 2 CaCl₂, and 2.0 MgSO₄ (pH 7.3). Both hippocampi were dissected out and sliced transversally (400 μm) using a tissue chopper with a cooled stage (custom-made by LIN, Magdeburg, Germany). The slices were maintained at room temperature in carbogen-bubbled ACSF (95% $O_2$/5% $CO_2$) containing 124 mM NaCl for a minimum of 2 h prior to the commencement of the recording. Recordings were performed in the same solution in a submerged chamber that was continuously superfused at 32 °C with carbogen-bubbled ACSF (1.2 ml/min). Recordings of field excitatory postsynaptic potentials (fEPSPs) were conducted in the *stratum radiatum* of the CA1b subfield with a glass pipette filled with ACSF. The resistance of the pipette was 1.4 MΩ. The stimulation pulses were applied to the Schaffer collaterals via a monopolar, electrolytically sharpened and lacquer-coated stainless-steel electrode, which was positioned approximately 300 μm closer to the CA3 subfield than the recording electrode. Basal synaptic transmission was monitored at a frequency of 0.05 Hz. Long-term potentiation was induced by applying five theta-burst stimulations (TBS) with an interval of 20 s. One TBS consisted of a single train of 10 bursts (4 pulses at 100 Hz) separated by 200 ms, with a duration of single pulses of 0.2 ms for each pulse. The stimulation strength was set to provide baseline fEPSPs with slopes of approximately 50% of the subthreshold maximum.

The data were recorded at a sampling rate of 10 kHz and then filtered between 0 and 5 kHz. They were subsequently analyzed using IntraCell software. The statistical evaluation of the mean changes in slope of fEPSPs during the final 10 min of recordings was conducted using the Student's t-test in SigmaPlot (Systat Software Inc., Chicago, IL, USA).

TIMP3 was dissolved in distilled water at a concentration of 5 μM and stored at −20 °C in small amounts. For each experiment, the final concentration of 15 nM was freshly prepared in ACSF and was bath-applied after 10 min stable baseline recording starting 30 min before TBS and ending 10 min after TBS stimulation. For control experiments the appropriate amount of bi-distilled water (vehicle) was applied. The data presented are from 10 slices/5 mice for the control group and 7 slices/4 mice for the TIMP3 group.

## Data availability

This study includes no data deposited in external repositories.

The source data of this paper are collected in the following database record: biostudies:S-SCDT-10_1038-S44319-025-00644-w.

## Peer review information

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

## Acknowledgements

We thank Kathrin Hartung and the team of the animal facility at the LIN Magdeburg for their excellent technical support throughout the project. We thank Eckart Gundelfinger and Helmut Brandstätter for their support and vivid discussions. This project was funded by German Research Foundation (DFG) FR 2758/3–1 and project 460333672 CRC1540 EBM (C02) to RF. Work in the authors CIS and AD labs is supported by the DFG 362321501/RTG 2413 SynAGE and 425899996/CRC1436.

## Author contributions

**Jeet Bahadur Singh**: Data curation; Formal analysis; Validation; Investigation; Visualization; Methodology; Writing—original draft. **Bartomeu Perelló-Amorós**: Data curation; Investigation. **Jenny Schneeberg**: Data curation; Investigation. **Hadi Mirzapourdelavar**: Resources; Data curation; Investigation; Writing—review and editing. **Constanze I Seidenbecher**: Resources; Funding acquisition; Writing—review and editing. **Anna Fejtová**: Conceptualization; Resources; Data curation; Formal analysis; Funding acquisition; Validation; Investigation; Methodology; Writing—review and editing. **Alexander Dityatev**: Conceptualization; Resources; Data curation; Formal analysis; Supervision; Funding acquisition; Validation; Investigation; Visualization; Methodology; Writing—original draft; Project administration; Writing—review and editing. **Renato Frischknecht**: Conceptualization; Resources; Data curation; Formal analysis; Supervision; Funding acquisition; Validation; Investigation; Visualization; Writing—original draft; Project administration; Writing—review and editing.

Source data underlying figure panels in this paper may have individual authorship assigned. Where available, figure panel/source data authorship is listed in the following database record: biostudies:S-SCDT-10_1038-S44319-025-00644-w.

## Funding

## Disclosure and competing interests statement

The authors declare no competing interests.

# Expanded View Figures

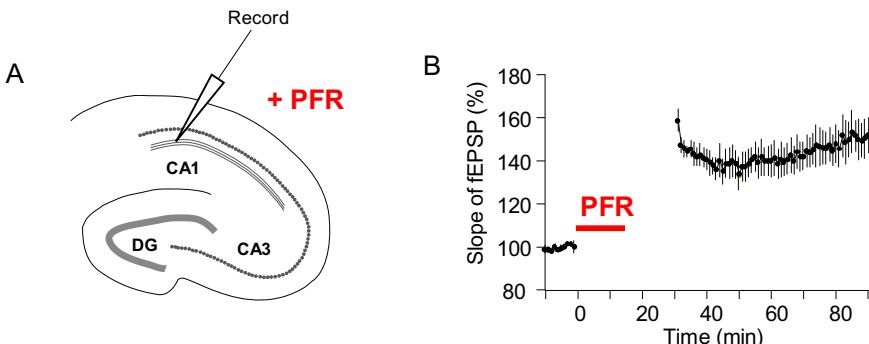

**Figure EV1. Extracellular electrophysiological recordings were made in the CA1 region of an acute hippocampal slice following PFR treatment.**

Note the robust and lasting increase in the slope of the field excitatory postsynaptic potential (fEPSP) 15 min after PFR treatment, which represents the induction of cLTP.

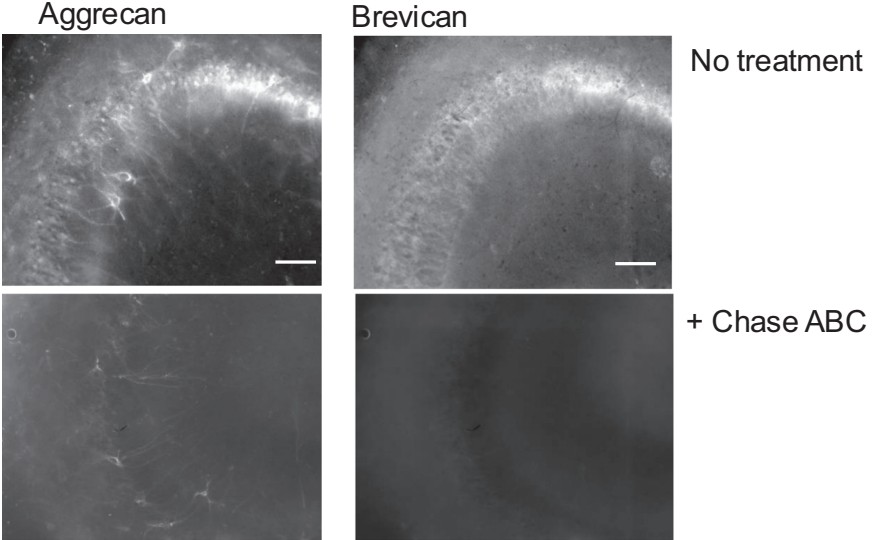

**Figure EV2. Chondroitinase ABC (Chase ABC) digestion abolishes ECM staining.**

Acute hippocampal slices from rats were stained with the ECM proteins aggrecan (left) and brevican (right). Incubation with chondroitinase ABC (lower panel) led to a strong reduction of aggrecan and brevican staining, indicating its solubilization (Scale bar 500 μm).

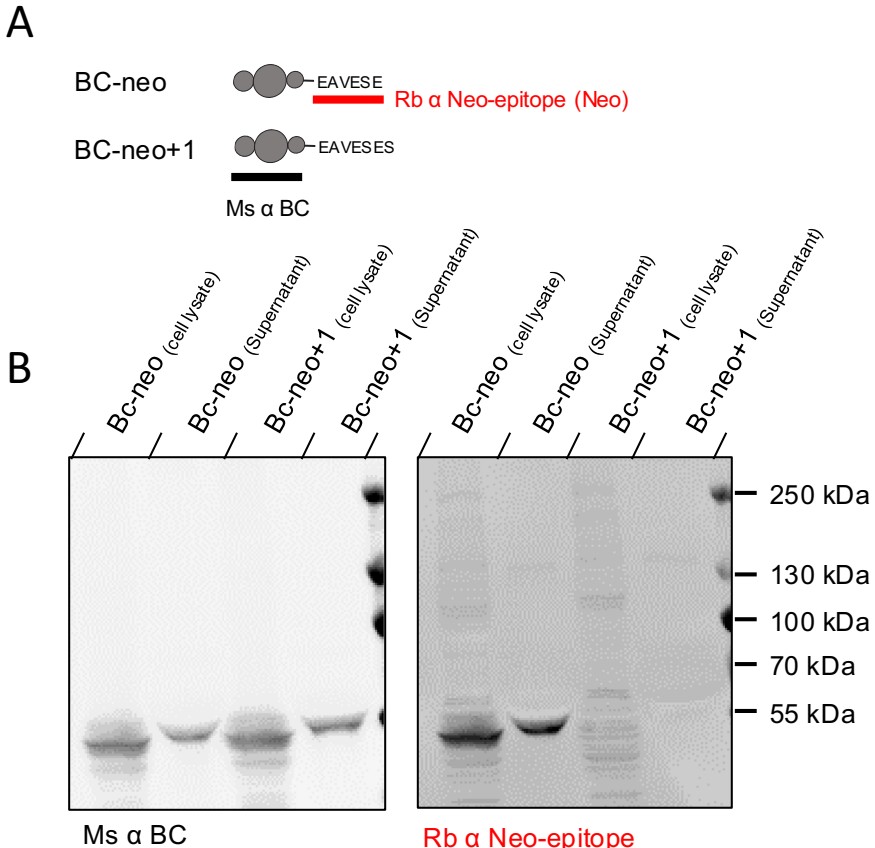

**Figure EV3. Neo-Epitope antibody detects exclusively proteolytically cleaved brevican between Glu$^{395}$-Ser$^{396}$.**

(A) Constructs overexpressed in HEK293T cells. Rat brevican construct representing the ADAMTS-derived N-terminal proteolytic fragment (BC-neo). Red underlined are the amino acids that served as epitope to generate the rb α Neo-epitope antibody. The C-terminus of BC-neo+1 includes the serine (S) that follows predicted ADAMTS cleavage site. (B) Western blots of Cell lysates and supernatants of HEK293T cells transfected with either BC-neo or BC-neo+1 constructs. Both constructs were detected in cell the lysate and supernatant using the Ms α BC antibody (left), which is not selective for the cleavage site. Rb α Neo antibody detected exclusively the BC-neo construct and failed to recognize BC-neo+1, confirming its high specificity for the ADAMTS-derived proteolytic fragment of brevican. Source data are available online for this figure.

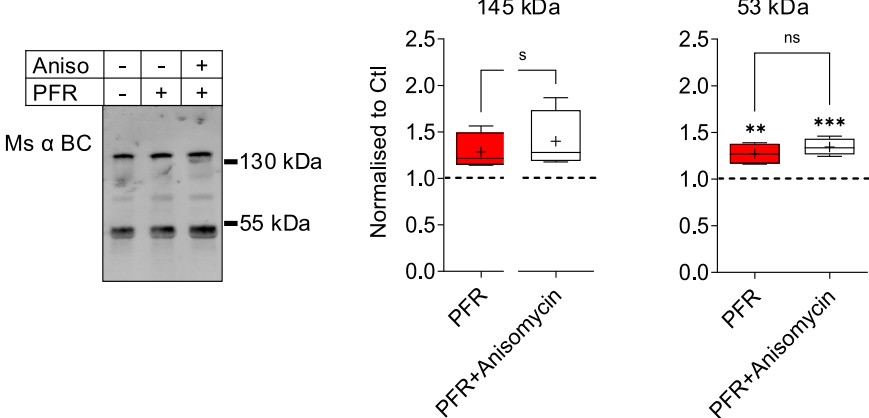

**Figure EV4. The protein synthesis inhibitor anisomycin did not affect the increase in brevican levels induced by activity.**

Hippocampal slices were pre-incubated with 20 μM anisomycin for 20–30 min prior to stimulation. Anisomycin had no effect on brevican full-length protein or brevican neo (53 kDa: Ctl vs. PFR: $P = 0.04$, Ctl vs. PFR+Anisomycin: $P < 0.001$. $n = 4$. One-way ANOVA, Šídák's multiple comparisons test, n.s. $P > 0.05$, *$P < 0.05$, **$P < 0.01$, ***$P < 0.001$. Box plot depicts the interquartile range (IQR, box), median is indicated as line, average as + and whiskers indicate minimal to maximal data point). For detailed statistics see Table EV8. Source data are available online for this figure.

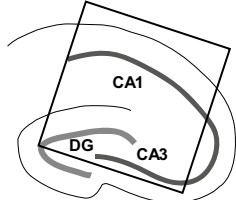

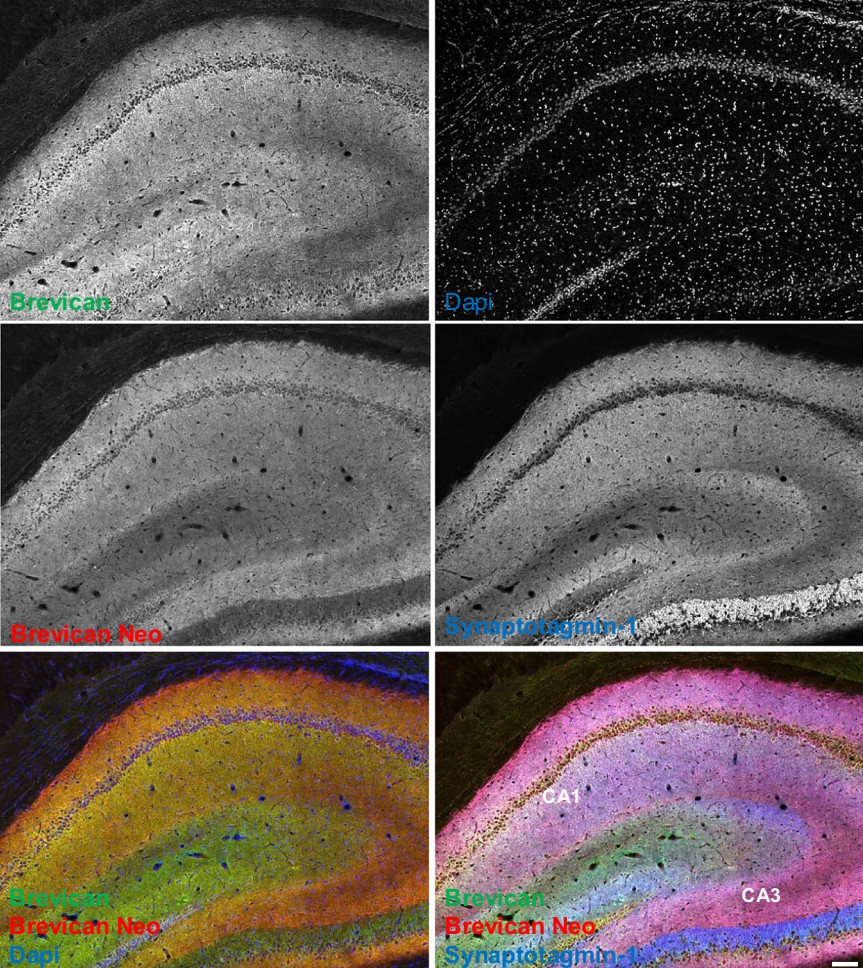

**Figure EV5. Distribution of brevican and brevican neo in the hippocampus.**

Cryosections of paraformaldehyde-fixed adult rat brains were stained for brevican using guinea pig anti-brevican (green; see also Valenzuela et al, 2014), brevican neo (red), synaptotagmin-1 (blue) antibody and DAPI. A tile scan of the hippocampus is depicted. Note the diffuse staining of both brevican and brevican neo throughout the hippocampus (scale bar 100 μm).

