## [Peer Review File · EMBO Reports]

Activity-dependent extracellular proteolytic cascade cleaves the ECM component brevican to promote structural plasticity

Jeet Singh, Bartomeu Perelló Amorós, Jenny Schneeberg, Hadi Mirzapourdelavar, Constanze Seidenbecher, Anna Fejtova, Alexander Dityatev, and Renato Frischknecht

Corresponding author(s): Renato Frischknecht (renato.frischknecht@fau.de)

Review Timeline:

Transfer Date:	3rd Dec 24
Editorial Decision:	6th Dec 24
Revision Received:	23rd Jun 25
Editorial Decision:	9th Oct 25
Revision Received:	20th Oct 25
Accepted:	4th Nov 25

Editor: Esther Schnapp

Transaction Report: This manuscript was transferred to EMBO reports following peer review at The EMBO Journal.

Referee #1:

The paper addresses an issue that is topical and important in neuroscience. The level of plasticity in the CNS is controlled by extracellular matrix molecules, particularly those associated with perineuronal nets. While the general principle of control of plasticity by proteoglycans is accepted, the mechanisms by which they affect individual synapses, and how these mechanisms are modified to allow or block synaptic changes are not established. The paper looks at one candidate CSPG, brevican and one candidate protease for degrading it, ADAMTS 4/5. The paper shows that brevican is degraded by this ADAMTS in slices during a form of LTP induced by a mixture of synaptically active molecules. The paper then goes on to implicate astrocytes, calcium channels associated with the postsynapse in the process, and shows that blocking ADAMTS activity inhibits changes in synaptic spines but does not affect LTP.

1) The paper has a single focus, brevican and its degradation. This is an advantage in that it simplifies the experiments and conclusions. However there are many proteoglycans and many proteases involved in controlling plasticity and none of this is mentioned. For instance ADAMTS4 has several targets in the ECM, and several proteases degrade CNS ECM, some of which eg MMP9 have an extensive literature related to synapses and plasticity. The role of brevican has been investigated in several ways, many of which are not mentioned. For instance brevican knockouts have normal LTP, while a multi-ECM molecule knockout has abnormal plasticity and memory. I can see why the authors would want to avoid all these complications, but at present the paper presents a very oversimplified view of a complex subject. At least the introduction and discussion need to be acknowledge that the issues are more complex than this set of experiments.

2) ADAMTS4/5 is a good candidate for activity-dependent modification of the CNS ECM, and it degrades several molecules. The paper would be stronger if degradation of a couple of further target molecules were investigated.

3) Immunostaining of brevican in the slices would show whether there is any change in its distribution during the LTP process.

4) The results on brevican and its products are all shown in relation to control. This does not give much idea of how much the actual quantities of brevican in the ECM are changed. The bands on the gels do not seem to change much. One would assume that a significant change in brevican in the region of synapses is required to change behavior during LTP. Can the authors address this?

Referee #2:

Considerable evidence much of it involving integrins indicates that adhesion chemistry is critically involved in synaptic plasticity (LTP) and possibly memory. These observations raise the strong possibility that production of the synaptic changes requires adjustments to the extracellular matrix (ECM). A number of studies have shown that experimental manipulations targeted at ECM constituents affect LTP and learning but detailed descriptions of matrix components at excitatory synapses in adult brain are lacking. The present manuscript tests the prediction that chemically induced synaptic potentiation ('cLTP') will cause rapid cleavage of the proteoglycan brevican, which is an important component of the ECM. The authors then describe pharmacological results that implicate a cell biological sequence in the observed proteolysis and convincingly demonstrate that cLTP is associated with an increase in spine-like filopodia. Intriguingly, blocking brevican cleavage does not disrupt physiological potentiation. The studies are described in a succinct and clear manner and the results are both novel and significant.

1) The presentation of the paper could be significantly improved by a clearer description of how the authors envision the layout of the brain ECM. There is a good likelihood that a 'loose' ECM surrounds neurons and their processes in the hippocampus and other forebrain structures, as described in the cited review by Fawcett et al (2022). A more specialized ECM referred to as perineuronal nets (PNN) is associated with a subpopulation of inhibitory interneurons and is thought, among other things, to associate with synapses on those cells. Other researchers hypothesize that ECM elements are found within or adjacent to the synaptic junction in areas in the adult forebrain and at that site to be critical for synaptic modifications (e.g., McGeachie et al, 2011). It would seem that the current authors assume that (a) the PNN arrangement is found at the base of hippocampal spines, and (b) this is the site for brevican cleavage during cLTP. Apparently, they do not envision a synaptic ECM enriched in brevican (and related lecticans). Information about the assumed distribution of the ECM within hippocampus would help readers evaluate the conclusions drawn by the authors.

2) Related to the above point, the manuscript does not do an adequate job of describing the literature on the localization of brevican in the hippocampus, particularly with regard to synapses. The authors would considerably improve the paper by showing that the brevican breakdown product in the hippocampus (Fig 1G) is concentrated at synapses. This would be a relatively straightforward dual immunohistochemical study.

3) It is also the case that the cited papers (Fawcett et al, Lupori et al.) do not provide strong evidence for the claim that ECM make-up, as seen in perineuronal nets, is found at all synapses.

4) The authors assert in the Introduction that LTP is accompanied by the appearance of

new spines. This is probably not the case for physiologically induced potentiation in the adult hippocampus -- light microscopic studies of spines and synapses do not report an increase in synapse number after theta burst induced LTP. A more conservative (and accurate) statement is that, as in the present study, cLTP increases the number of spine-like structures on dendrites.

5) It is not clear if the authors measured synaptic responses during the application of the various antagonists and D-Serine. This is an important issue because induced pathophysiology could interfere with brevicin proteolysis independently of the targeted cellular process (e.g., ADAMTS activity). The paper needs to describe the physiological consequences (if any) of the pharmacological treatments.

6) A panel illustrating the time course and magnitude of the potentiation produced by the cLTP treatment (picrotoxin, forskolin, and Rolipram) should be added to Figure 1. The authors should also test if the potentiation produced by the cLTP protocol is occluded by prior induction of LTP with theta burst stimulation.

7) The photomicrograph in Fig 1G is not acceptable. The summary graph to the right shows that the experimental treatment produced a small but statistically robust increase in fluorescence. The images describe a massive difference between the two groups -this example cannot be typical. It would also be helpful to include a higher resolution image so that readers have a better idea of what is labeled by the antibody.

Referee #3:

The dynamic remodeling of extracellular matrix (ECM) proteins plays a key role in the formation and plasticity of central synapses. In this manuscript by Singh et al., the regulation of proteolytic cleavage of brevicin, a chondroitin sulfate proteoglycan in the perineuronal ECM, by ADAMTS proteases during long-term potentiation (LTP) in hippocampal synapses was investigated. The authors showed that (1) chemical LTP (cLTP) induction increases the amount of cleaved brevicin in the supernatant samples of rat hippocampal slices; (2) ADAMTS proteases are involved in regulating brevicin cleavage, depending on the activity of NMDA receptors, voltage-gated calcium channels, and CaMKII; (3) inhibition of ADAMTS activity showed no effects on early LTP functionally but affected the structural changes in dendritic protrusions.

While this research topic is potentially interesting for understanding the molecular mechanisms underlying activity-dependent ECM remodeling during structural plasticity in central synapses, I have several major concerns regarding the experimental methodologies

and result interpretations, which greatly dampened my enthusiasm for this work. Therefore, I feel that this work is still preliminary for consideration for publication in EMBO J.

1. The study mainly used biochemical analyses in combination with pharmacological treatments to show the molecular regulation of the proteolytic cleavage of brevicin induced by cLTP. Due to concerns about the specificity and effectiveness of the pharmacological inhibitors used, I would expect to see molecular manipulation on the expression and/or activity of the key regulators involved (e.g., ADAMTS, GRIN2B, VGCC, and CaMKII) to validate the findings.
2. The immunostaining data in Fig. 1G showed a slight but significant increase in Neo intensity in the CA1 regions in response to cLTP treatment. What about the other regions where brevicin is minimal, as indicated in Suppl Fig 1? Additionally, Neo immunostaining data should be provided to support the Western blot data in the subsequent pharmacological experiments to delineate the molecular regulation of cLTP-induced brevicin cleavage (Figs. 2-4).
3. Given the large variations in the density and morphology of dendritic spines in the hippocampus, the imaging studies that aim to show the involvement of TIMP3 in cLTP-induced formation of dendritic spines (Fig. 6) would be more convincing if they looked at the same regions of dendrites in acute hippocampal slices before and after the treatment.
4. Fig. 3A: The representative blot is of poor quality. Additionally, the difference in Neo band intensity reflected in the blot seems contradictory to the quantitative results shown in the graph.

Dear Renato,

Thank you for the transfer of your manuscript to EMBO reports.

As we discussed, I would like to invite you to revise your manuscript with the understanding that the referee concerns must be fully addressed and their suggestions taken on board, except for point 1 by referee 3, which does not need to be addressed experimentally.

Please address all referee concerns in a complete point-by-point response. Acceptance of the manuscript will depend on a positive outcome of a second round of review. It is EMBO reports policy to allow a single round of major revision only and acceptance or rejection of the manuscript will therefore depend on the completeness of your responses included in the next, final version of the manuscript.

We realize that it is difficult to revise to a specific deadline. In the interest of protecting the conceptual advance provided by the work, we recommend a revision within 3 months (8th Mar 2025). Please discuss the revision progress ahead of this time with the editor if you require more time to complete the revisions.

- 1) A data availability section providing access to data deposited in public databases is missing. If you have not deposited any data, please add a sentence to the data availability section that explains that.
- 2) Your manuscript contains statistics and error bars based on $n=2$. Please use scatter blots in these cases. No statistics should be calculated if $n=2$.

5) a complete author checklist, which you can download from our author guidelines <https://www.embopress.org/page/journal/14693178/authorguide>. Please insert information in the checklist that is also reflected in the manuscript. The completed author checklist will also be part of the RPF.

6) Please note that all corresponding authors are required to supply an ORCID ID for their name upon submission of a revised manuscript (<https://orcid.org/>). Please find instructions on how to link your ORCID ID to your account in our manuscript tracking system in our Author guidelines <https://www.embopress.org/page/journal/14693178/authorguide#authorshipguidelines>

7) Before submitting your revision, primary datasets produced in this study need to be deposited in an appropriate public database (see <https://www.embopress.org/page/journal/14693178/authorguide#datadeposition>). Please remember to provide a reviewer password if the datasets are not yet public. The accession numbers and database should be listed in a formal "Data

Availability" section placed after Materials & Method (see also <https://www.embopress.org/page/journal/14693178/authorguide#datadeposition>). Please note that the Data Availability Section is restricted to new primary data that are part of this study. * Note - All links should resolve to a page where the data can be accessed. *

12) All Materials and Methods need to be described in the main text using our 'Structured Methods' format, which is required for all research articles. According to this format, the Methods section includes a separate Reagents and Tools Table file (listing key reagents, experimental models, software and relevant equipment and including their sources and relevant identifiers) and a Methods and Protocols section describing the methods using a step-by-step protocol format. The aim is to facilitate adoption of the methodologies across labs. More information on how to adhere to this format as well as a downloadable template (.docx) for the Reagents and Tools Table can be found in our author guidelines:

An example of a Method paper with Structured Methods can be found here: <https://www.embopress.org/doi/full/10.1038/s44320-024-00037-6#sec-4>

You are able to opt out of this by letting the editorial office know (emboreports@embo.org). If you do opt out, the Review Process File link will point to the following statement: "No Review Process File is available with this article, as the authors have

chosen not to make the review process public in this case."

I look forward to seeing a revised form of your manuscript when it is ready. Please use this link to submit your revision:
<https://embor.msubmit.net/cgi-bin/main.plex>

Point by point answer to the reviewers:

Referee

#1:

The paper addresses an issue that is topical and important in neuroscience. The level of plasticity in the CNS is controlled by extracellular matrix molecules, particularly those associated with perineuronal nets. While the general principle of control of plasticity by proteoglycans is accepted, the mechanisms by which they affect individual synapses, and how these mechanisms are modified to allow or block synaptic changes are not established. The paper looks at one candidate CSPG, brevican and one candidate protease for degrading it, ADAMTS 4/5. The paper shows that brevican is degraded by this ADAMTS in slices during a form of LTP induced by a mixture of synaptically active molecules. The paper then goes on to implicate astrocytes, calcium channels associated with the postsynapse in the process, and shows that blocking ADAMTS activity inhibits changes in synaptic spines but does not affect LTP.

1) The paper has a single focus, brevican and its degradation. This is an advantage in that it simplifies the experiments and conclusions. However, there are many proteoglycans and many proteases involved in controlling plasticity and none of this is mentioned. For instance ADAMTS4 has several targets in the ECM, and several proteases degrade CNS ECM, some of which eg MMP9 have an extensive literature related to synapses and plasticity.

The role of brevican has been investigated in several ways, many of which are not mentioned.

For instance **brevican knockouts** have normal LTP, while a multi-ECM molecule knockout has abnormal plasticity and memory. I can see why the authors would want to avoid all these complications, but at present the paper presents a very oversimplified view of a complex subject. At least the introduction and discussion need to be acknowledge that the issues are more complex than this set of experiments.

We have indeed focused on brevican because it is highly abundant within the loose ECM and because the antibody recognizing the specific ADAMTS-derived cleavage product exists and is established for analysis on Western blots as well as on immunostainings on brain slices. According to the reviewer's request, we now added new text to describe the ECM components, ECM proteolysis and their effects on neuroplasticity in introduction and discussion. We also discussed, how the observed defects in structural plasticity but normal induction of early LTP align with previously published defects in LTP in brevican KO animals (Brakebusch et al, 2002).

2) ADAMTS4/5 is a good candidate for activity-dependent modification of the CNS ECM, and it degrades several molecules. The paper would be stronger if degradation of a couple of further target molecules were investigated.

Indeed, ADAMTS4/5 has been described to cleave brevican and other lecticans like aggrecan, versican in the cartilage and possibly also neurocan. It also processes other substrates including cartilage oligomeric matrix protein and matrilin-3, which are however not present within the perisynaptic ECM.

There is no specific antibody that would recognize the hypothetical ADAMTS4/5-specific cleavage fragment of neurocan and versican was not detectable in brain samples in our hands. A specific neo antibody detecting the new C-terminus emerging after ADAMTS cleavage exists for aggrecan but works only for Western blotting in our hands. We have now included Western blot using this antibody in Figure 1 showing cLTP-induced cleavage of aggrecan.

3) Immunostating of brevican in the slices would show whether there is any change in its distribution during the LTP process.

To address this important issue, we analyzed changes abundance of neo epitope within the loose ECM and specifically around synapses upon cLTP in acute hippocampal slices and observed cLTP-induced increase. These results are now included in Figure 6F-J.

4) The results on brevican and its products are all shown in relation to control. This does not give much idea of how much the actual quantities of brevican in the ECM are changed. The bands on the gels do not seem to change much. One would assume that a significant change in brevican in the region of synapses is required to change behavior during LTP. Can the authors address this?

In the revised version, we included the iconographic to explain experimental strategy and timeline experiment that shows brevican processing at different timepoints after cLTP induction. In each experiment, we included Ctl and PFR-treated slices from the same animals. We normalized all blots for total blotted protein using the stain-free method to correct for variability caused by protein loading and the blotting procedure. Using this methodology, we measured an increase in brevican cleavage ranging from 15-50% in some experiments, and up to 100% relative to untreated slices. Indeed, all our blots are semi-quantitative and our data gives no information of molar content of the proteins detected. We are not able to assign molarity to our experiments retrospectively. However, the changes in the protein abundance were 15 -100%, which is in our view biologically relevant, and effects were highly statistically significant. In order to be more correct to this fact, we now avoid the term “quantitative Western blot” and replaced it with “semi-quantitative”.

Referee

#2:

Considerable evidence much of it involving integrins indicates that adhesion chemistry is critically involved in synaptic plasticity (LTP) and possibly memory. These observations raise the strong possibility that production of the synaptic changes requires adjustments to the extracellular matrix (ECM). A number of studies have shown that experimental manipulations targeted at ECM constituents affect LTP and learning but detailed descriptions of matrix components at excitatory synapses in adult brain are lacking. The present manuscript tests the prediction that chemically induced synaptic potentiation ('cLTP') will cause rapid cleavage of the proteoglycan brevican, which is an important component of the ECM. The authors then describe pharmacological results that implicate a cell biological sequence in the observed proteolysis and convincingly demonstrate that cLTP is associated with an increase in spine-like filopodia. Intriguingly, blocking brevican

cleavage does not disrupt physiological potentiation. The studies are described in a succinct and clear manner and the results are both novel and significant.

1) The presentation of the paper could be significantly improved by a clearer description of how the authors envision the layout of the brain ECM.

There is a good likelihood that a 'loose' ECM surrounds neurons and their processes in the hippocampus and other forebrain structures, as described in the cited review by Fawcett et al (2022). A more specialized ECM referred to as perineuronal nets (PNN) is associated with a subpopulation of inhibitory interneurons and is thought, among other things, to associate with synapses on those cells. Other researchers hypothesize that ECM elements are found within or adjacent to the synaptic junction in areas in the adult forebrain and at that site to be critical for synaptic modifications (e.g., McGeachie et al, 2011). It would seem that the current authors assume that (a) the PNN arrangement is found at the base of hippocampal spines, and (b) this is the site for brevicin cleavage during cLTP. Apparently, they do not envision a synaptic ECM enriched in brevicin (and related lecticans). Information about the assumed distribution of the ECM within hippocampus would help readers evaluate the conclusions drawn by the authors.

This is a very important point and we addressed it intensively in the revised version of our article.

Indeed, we share the view of two forms of ECM and we think that in the adult hippocampus (and cortex) virtually all neurons are surrounded by the loose ECM. Most synapses are embedded in ECM, however ECM is likely not present within the synaptic cleft as it is described for the neuromuscular junction. Our previous work showing that ECM removal altered lateral diffusion of AMPA receptor in the membrane of excitatory hippocampal neurons but not within the synaptic cleft supports this view (Frischknecht et al, 2009).

To make this point clearer, we provide more precise and structured background information in the introduction of the revised manuscript. Specifically, we described the perisynaptic loose ECM and PNN as two complementary forms of ECM with specific molecular composition and functions. Further, we included the completely new figure 6 showing 1) the localization of brevicin and aggrecan to PNN and loose ECM within hippocampus, 2) specific localization of brevicin and its ADAMTS4/5-dependent fragment to the perisynaptic locations, where they align but not colocalize with synaptic markers and 3) probably most importantly, the cLTP-induced generation of brevicin fragment specifically in the perisynaptic regions. We also adjusted the discussion accordingly.

2) Related to the above point, the manuscript does not do an adequate job of describing the literature on the localization of brevicin in the hippocampus, particularly with regard to synapses. The authors would considerably improve the paper by showing that the brevicin breakdown product in the hippocampus (Fig 1G) is concentrated at synapses. This would be a relatively straightforward dual immunohistochemical study.

As already described in the answer to previous point, we have rewritten large sections of the introduction to better introduce the localization of ECM and implemented a paragraph about ECM localization in relation to synapses in the section "Discussion". Further we added immunofluorescence images showing brevicin and its N-terminal proteolytic fragment in the hippocampal CA1 of adult rats together with the synaptic marker protein

bassoon and Homer1 as requested (Figure 6). Further, we have quantified brevican cleavage at synapses to assess synaptic and perisynaptic proteolysis (Figure 6).

3) It is also the case that the cited papers (Fawcett et al, Lupori et al.) do not provide strong evidence for the claim that ECM make-up, as seen in perineuronal nets, is found at all synapses.

This valid comment was addressed by changes in the introduction, addition of new results (Figure 6 and expanded view Figure EV5) and adjusted discussion that are in detail described in our answers to the point 2 and 3 above.

4) The authors assert in the Introduction that LTP is accompanied by the appearance of new spines. This is probably not the case for physiologically induced potentiation in the adult hippocampus -- light microscopic studies of spines and synapses do not report an increase in synapse number after theta burst induced LTP. A more conservative (and accurate) statement is that, as in the present study, cLTP increases the number of spine-like structures on dendrites.

We agree that in the original version, our wording was not completely clear. We meant not increased synapse numbers but increased spine dynamics, which was in our study quantified as number of dendritic protrusions. We changed the wording accordingly and added references explaining the link between spine dynamics and structural plasticity, which is needed for stable LTP and engram formation.

5) *It is not clear if the authors measured synaptic responses during the application of the various antagonists and D-Serine. This is an important issue because induced pathophysiology could interfere with brevican proteolysis independently of the targeted cellular process (e.g., ADAMTS activity). The paper needs to describe the physiological consequences (if any) of the pharmacological treatments.*

This question regards experiments in Figure 4, in which we investigated the role of astroglia in brevican processing. To do so we have used two compounds to interfere with glial function: carbenoxolone and endothelin-1. These compounds have been used in several papers spanning over two decades. As requested by the reviewer, we have included more information about the function and physiological effects of the compounds in the results section and added new citations in the revised manuscript.

In our experiments, both compounds blocked brevican processing, indicating at first glance a role of glia cells in the process. However, brevican cleavage was completely rescued by co-application of the inhibitors with D-serine, showing that the drugs do not have unspecific pathological effect, but rather that D-serine signaling is necessary for cLTP-induced brevican cleavage by ADAMTS4/5.

6) A panel illustrating the time course and magnitude of the potentiation produced by the cLTP treatment (picrotoxin, forskolin, and Rolipram) should be added to Figure 1. The authors should also test if the potentiation produced by the cLTP protocol is occluded by prior induction of LTP with theta burst stimulation.

As requested by reviewer, we have now included fEPSP recording, showing a robust potentiation after PFR treatment in Figure EV1.

The second point of the reviewer addresses the mechanisms of LTP induction by PFR. PFR treatment has been extensively characterized by Lisman lab as a method producing NMDA-dependent LTP in large number of synapses of CA3-CA1 Schaffer collateral pathway (Otmakhov et al, 2004a; Otmakhov et al, 2004b). Since then, it has been used in multiple studies to investigate e.g. spine dynamics, AMPA receptor trafficking and extracellular proteolysis (e.g. (Kopeck et al, 2006; Matsumoto-Miyai et al, 2009). We have added citations of these studies, some of which also include occlusion experiments. The goal of our work was to investigate brevicin cleavage in condition of NMDA-dependent LTP induction in CA3-CA1 connections, for which we are convinced this treatment is well suited. Studies of possible occlusion of cLTP induction by prior LTP induction by theta burst stimulation are beyond the scope of this work.

7) The photomicrograph in **Fig 1G** is not acceptable. The summary graph to the right shows that the experimental treatment produced a small but statistically robust increase in fluorescence. The images describe a massive difference between the two groups -this example cannot be typical. It would also be helpful to include a higher resolution image so that readers have a better idea of what is labeled by the antibody.

We replaced the example with a more representative image. Additionally, we have included high-resolution images in Figure 2 and overview images in Supplementary Figure 6 to provide a clearer view of the staining quality.

Referee**#3:**

The dynamic remodeling of extracellular matrix (ECM) proteins plays a key role in the formation and plasticity of central synapses. In this manuscript by Singh et al., the regulation of proteolytic cleavage of brevicin, a chondroitin sulfate proteoglycan in the perineuronal ECM, by ADAMTS proteases during long-term potentiation (LTP) in hippocampal synapses was investigated. The authors showed that (1) chemical LTP (cLTP) induction increases the amount of cleaved brevicin in the supernatant samples of rat hippocampal slices; (2) ADAMTS proteases are involved in regulating brevicin cleavage, depending on the activity of NMDA receptors, voltage-gated calcium channels, and CaMKII; (3) inhibition of ADAMTS activity showed no effects on early LTP functionally but affected the structural changes in dendritic protrusions.

While this research topic is potentially interesting for understanding the molecular mechanisms underlying activity-dependent ECM remodeling during structural plasticity in central synapses, I have several major concerns regarding the experimental methodologies and result interpretations, which greatly dampened my enthusiasm for this work. Therefore, I feel that this work is still preliminary for consideration for publication in EMBO

J.

1. The study mainly used biochemical analyses in combination with pharmacological treatments to show the molecular regulation of the proteolytic cleavage of brevicin induced by cLTP. Due to concerns about the specificity and effectiveness of the pharmacological inhibitors used, I would expect to see molecular manipulation on the

expression and/or activity of the key regulators involved (e.g., ADAMTS, GRIN2B, VGCC, and CaMKII) to validate the findings.

The focus of our work was to study mechanisms of activity-dependent proteolysis of brevicin and its effect on neuronal plasticity. For that purpose, we have used a number of pharmacological treatments to block various proteases, Ca²⁺ channels and neurotransmitter receptors. The compounds used in our study have been used extensively past decades and are very well characterized; this is especially true for glutamate receptor antagonists and VGCC blocker such as CNQX and nifedipine, respectively. However, we do agree that concerns about the specificity and effectiveness in experiments are always justified. Therefore, to minimize off-target effects of used drugs, we have applied multiple inhibitors targeting the same process in nearly all experiments. We have used three different metalloprotease inhibitors and furin-like protease inhibitors, two drugs interfering with astrocyte-dependent D-serine signaling. NMDA receptor activity was blocked by broadly used NMDA receptors antagonists and D-serine. Thus, we are very confident about specificity of the treatments and validity of our conclusions. Although studies of genetic models e.g. ko mice of the ADAMTS, GRIN2B VGCC or CaMKII would be interesting, this goes far beyond the scope of our work.

2. The immunostaining data in Fig. 1G showed a slight but significant increase in Neo intensity in the CA1 regions in response to cLTP treatment. **What about the other regions where brevicin is minimal, as indicated in Suppl Fig 1?** Additionally, Neo immunostaining data should be provided to support the Western blot data in the subsequent pharmacological experiments to delineate the molecular regulation of cLTP-induced brevicin cleavage (Figs. 2-4).

The supplemental figure 1 in the original submission actually showed loss of brevicin immunoreactivity upon extraction of ECM from acute slices and was intended to document that the enzymatic extraction of ECM is suitable step to prepare samples for Western blot analyses. All animals that we tested show brevicin staining in the loose ECM throughout the hippocampus and cortex. We improved description of experimental procedure and additionally added entire figure showing the distribution of brevicin and its ADAMTS4/5 cleavage product in the hippocampus and cortex and the cLTP-induced cleavage of brevicin in perisynaptic regions of the hippocampus.

We agree that analysis of immunostainings for all treatments may potentially reveal additional location-specific regulation of brevicin proteolysis. However, backing each treatment with quantitative immunostaining and meaningful statistical results, would require experiments on an additional 40–50 rats. In our opinion, the potential gain in information does not justify the usage of so many additional animals.

3. Given the large variations in the density and morphology of dendritic spines in the hippocampus, the imaging studies that aim to show the involvement of TIMP3 in cLTP-induced formation of dendritic spines (Fig. 6) would be more convincing if they looked at the same regions of dendrites in acute hippocampal slices before and after the treatment.

We were aware of the large biological variability of dendritic morphology and therefore we were very stringent in the planning, conducting and analysis of these experiments.

We have used mice that express yellow fluorescent protein (YFP) in a sparse population of neurons in the CA1 region and in granular cells of the dentate gyrus (Figure 7). We analyzed only the secondary branches of the apical dendrites of CA1 pyramidal cells with similar diameters located 100 μm away from the pyramidal layer. Indeed, there was significant variation in the number of spines per 10 μm under control and treated conditions. Nevertheless, the cLTP-induced effects (increase in dendritic protrusions) that we observed after PFR treatment were highly significant with this experimental setting and similar to previously published data (Matsumoto-Miyai *et al.*, 2009). Similarly, the treatment with TIMP3 completely disrupted the effect of cLTP. The statistic effects were highly significant (one way ANOVA, Dunnett's multiple comparison test, $P < 0.01$ and $P < 0.001$), which underscores the robustness of the effect.

Thus, we observed robust, biologically meaningful and statistically highly significant changes. Therefore, we do see any justification for repetition of experiments with experimental setup, which is potentially more sensitive. Moreover, it would be difficult to justify usage of additional set of animals for this purpose.

4. Fig. 3A: The representative blot is of poor quality. Additionally, the difference in Neo band intensity reflected in the blot seems contradictory to the quantitative results shown in the graph.

We replaced the blot by a qualitatively better one, as requested.

References

- Brakebusch C, Seidenbecher CI, Asztely F, Rauch U, Matthies H, Meyer H, Krug M, Bockers TM, Zhou X, Kreutz MR *et al* (2002) Brevican-deficient mice display impaired hippocampal CA1 long-term potentiation but show no obvious deficits in learning and memory. *Mol Cell Biol* 22: 7417-7427
- Frischknecht R, Heine M, Perrais D, Seidenbecher CI, Choquet D, Gundelfinger ED (2009) Brain extracellular matrix affects AMPA receptor lateral mobility and short-term synaptic plasticity. *Nat Neurosci* 12: 897-904
- Kopec CD, Li B, Wei W, Boehm J, Malinow R (2006) Glutamate receptor exocytosis and spine enlargement during chemically induced long-term potentiation. *J Neurosci* 26: 2000-2009
- Matsumoto-Miyai K, Sokolowska E, Zurlinden A, Gee CE, Luscher D, Hettwer S, Wolfel J, Ladner AP, Ster J, Gerber U *et al* (2009) Coincident pre- and postsynaptic activation induces dendritic filopodia via neurotrypsin-dependent agrin cleavage. *Cell* 136: 1161-1171
- Otmakhov N, Khibnik L, Otmakhova N, Carpenter S, Riahi S, Asrican B, Lisman J (2004a) Forskolol-induced LTP in the CA1 hippocampal region is NMDA receptor dependent. *J Neurophysiol* 91: 1955-1962
- Otmakhov N, Tao-Cheng JH, Carpenter S, Asrican B, Dosemeci A, Reese TS, Lisman J (2004b) Persistent accumulation of calcium/calmodulin-dependent protein kinase II in dendritic spines after induction of NMDA receptor-dependent chemical long-term potentiation. *J Neurosci* 24: 9324-9331

Dear Renato,

Thank you for the submission of your revised manuscript. We have now received the enclosed reports from referees 1 and 3 as well as cross-comments from referee 1. Referee 2 was unfortunately not responsive to our queries, and I am therefore making a decision now based on the 2 reports we have.

As you will see, while referee 3 still has concerns with the revised study, referee 1 is satisfied; although s/he agrees that the points raised by referee 3 are valid. I would like to invite you to address the last referee comments in a point-by-point response to be submitted with your final ms and that will be published as part of your transparent peer-review process file.

A few editorial requests will also need to be addressed before we can proceed with the official acceptance of your manuscript:

- Please add up to 5 keywords to your ms file.
- Please add a "Disclosure and Competing Interests" statement to your ms file.
- The author credits need to be removed from the ms file. All credits need to be entered during online ms submission.
- In the author checklist several questions have not been answered. Please choose for all questions one of the answers from the pulldown menu and send us a new list with your final ms.
- The suppl. figures are uploaded in one PDF file. This file should either be called Appendix or you could upload the 5 figures separately as EV figures (Figures EV1-EV5 whose legends would be in the ms file). EV figures are embedded in the ms text online. Please see our guide to authors for more information. Please correct the file names and all callouts in the ms text.
- The Expanded view tables can also be uploaded as separate Tables EV1, etc. Or as one excel file called Table EV1 with several tabs.
- Callouts are missing for Figure 1K, Table 1 (Table EV1 is called out but missing); table EV6 is called out but missing, please correct.
- The Methods section should include a separate Reagents and Tools Table file (listing key reagents, experimental models, software and relevant equipment and including their sources and relevant identifiers) and a Methods and Protocols section in which the authors should describe their methods using a step-by-step protocol format with bullet points, to facilitate the adoption of the methodologies across labs. More information on how to adhere to this format as well as downloadable templates (.docx) for the Reagents and Tools Table can be found in our author guidelines: <<https://www.embopress.org/page/journal/14693178/authorguide#manuscriptpreparation>>.
- The Source Data for Figure 6 and 7 should be part of their respective folders; all SD need to be grouped and uploaded as one folder per figure. SD for 6D, 6E seem to be missing, please correct.
- The Figure Legends should be moved to after the References.
- I understand that Christopher contacted you regarding the data in Fig 1i and their source data. Please send us with your final ms the corrected figure panel and SD.

* Figure legends *

1. Please note that the legend for figures 1K, 3C, D; 4C, E, F; 5C, D, E is missing in the manuscript. This needs to be rectified.
2. Please note that the exact p values are not provided in the legends of figures 1F, G, H, K; 2B, C; 3B, D; 4B, C, E, F; 5B, C; 6H, J; 7F, G. Please provide exact values as reasonable.
3. Please note that the box plots need to be defined in terms of minima, maxima, centre, bounds of box and whiskers, and percentile in the legends of figures 1F, G, H, K; 2B, C; 3B, D; 4B, C, E, F; 5B, C; 6H, J; 7F, G; S4
4. Please note that information related to n is missing in the legends of figures 1D, K

I would like to suggest a few minor changes to the abstract that needs to be written in present tense. Do you agree with this:

The brain's perineuronal extracellular matrix (ECM) is a crucial factor in maintaining the stability of mature brain circuitry. However, how activity-induced synaptic plasticity is achieved in the adult brain with a dense ECM is unclear. We hypothesized that neuronal activity induces cleavage of ECM, creating conditions for synaptic rearrangements. To test this hypothesis, we

investigated neuronal activity-dependent proteolytic cleavage of brevican, a prototypical ECM proteoglycan, and the importance of this process for functional and structural synaptic plasticity in the rat hippocampus *ex vivo*. Our findings reveal that chemical long-term potentiation (cLTP) triggers rapid brevican cleavage in perisynaptic regions through the activation of an extracellular proteolytic cascade involving proprotein convertases and ADAMTS-4 and ADAMTS-5. This process requires NMDA receptor activation and involves astrocytes. Interfering with cLTP-induced brevican cleavage prevents the formation of new dendritic protrusions in CA1 but does not impact LTP induction by theta-burst stimulation of CA3-CA1 synapses. Our data reveal a mechanism of activity-dependent ECM remodeling and suggest that ECM degradation is essential for structural synaptic plasticity.

The synopsis image content is OK but the text is too small at the final image size of 550 pixels wide. Please send us a new synopsis image at its final size with larger text. We also need A) a short (1-2 sentences) summary of the findings and their significance, and B) 2-3 bullet points highlighting key results.

I look forward to seeing a final version of your manuscript as soon as possible. Please use this link to submit your revision: <https://embor.msubmit.net/cgi-bin/main.plex>

Referee #1:

The authors have responded appropriately to the comments of the reviewers. I do not think that a further round of revisions will improve the manuscript. I suggest publication of the submitted manuscript.

Referee #3:

In this revised manuscript, it is disappointing to see that my major concerns remain unaddressed. While the author's response reiterates that the study focuses on elucidating the mechanisms of activity-dependent proteolysis of brevican and its effects on neuronal plasticity, the issue of inhibitor specificity as raised in the previous round of review remains a central concern.

It is entirely appropriate to use well-characterized pharmacological agents that have been employed for decades, such as CNQX and nifedipine, for glutamate receptor and VGCC inhibition, as stated. However, the conclusions drawn from the use of TIMP-3 and piceatannol to suggest specific involvement of ADAMTS are problematic. A substantial body of literature indicates that these two inhibitors also affect the activity of various members of the MMP family. Furthermore, the author claims that piceatannol is an ADAMTS-4 inhibitor, citing Lauer-Fields et al., 2008 on p. 6, yet this reference appears to be missing from the reference list. Therefore, I continue to believe that molecular manipulation of key molecules such as ADAMTS-4 (such as using approaches like AAV-mediated CRISPRi or RNAi-based gene silencing, rather than genetic manipulation via knockout animals as mentioned in the response letter) is essential to validate the findings.

Additionally, concerns regarding the substantial variation in dendritic spine density and morphology persist. While it is noted that small p-values were obtained through careful planning and analysis (Figure 7F and G), the changes in the values of dendritic protrusion length and density between control and treated groups remain very small and subtle. I appreciate the authors' intention to reduce the number of experimental animals to be used for additional experiments in the revised submission. Comparing the same regions of dendrites in hippocampal slices before and after treatment would provide more compelling evidence without requiring a large sample size.

Cross-comments from referee 1:

The comments from referee 3 are valid. However my view was that the paper, although imperfect, has sufficient interest and solid results that it should be published now rather than waiting for further tinkering to make it perfect.

Point by point answer to the reviewers:

Referee**#1:**

The authors have responded appropriately to the comments of the reviewers. I do not think that a further round of revisions will improve the manuscript. I suggest publication of the submitted manuscript.

We would like to thank the referee for the constructive feedback, which we believe has helped us to substantially improve our manuscript.

Referee**#3:**

In this revised manuscript, it is disappointing to see that my major concerns remain unaddressed. While the author's response reiterates that the study focuses on elucidating the mechanisms of activity-dependent proteolysis of brevican and its effects on neuronal plasticity, the issue of inhibitor specificity as raised in the previous round of review remains a central concern.

It is entirely appropriate to use well-characterized pharmacological agents that have been employed for decades, such as CNQX and nifedipine, for glutamate receptor and VGCC inhibition, as stated. However, the conclusions drawn from the use of TIMP-3 and piceatannol to suggest specific involvement of ADAMTS are problematic. A substantial body of literature indicates that these two inhibitors also affect the activity of various members of the MMP family.

We agree that inhibitor specificity is an important issue. However, our conclusion that proteases from the ADAMTS family are involved in this process is not based solely on data obtained using protease inhibitors, which as reviewer correctly states have broader specificity. Our conclusion is also based on the monitoring of the specific cleavage of brevican between the amino acids Glu³⁹⁵-Ser³⁹⁶ detected by the specific Neo antibody. Nakamura and coauthors demonstrated that cleavage of brevican at this site is exerted by proteases from the ADAMTS family (Nakamura et al. 2000). Thus, based on these two independent evidences we claim that proteases of ADAMTS family are involved.

Furthermore, the author claims that piceatannol is an ADAMTS-4 inhibitor, citing Lauer-Fields et al., 2008 on p. 6, yet this reference appears to be missing from the reference list.

We apologize for the missing reference and have now added it to the list.

Therefore, I continue to believe that molecular manipulation of key molecules such as ADAMTS-4 (such as using approaches like AAV-mediated CRISPRi or RNAi-based gene silencing, rather than genetic manipulation via knockout animals as mentioned in the response letter) is essential to validate the findings.

We agree that in vivo AAV-mediated CRISPRi or RNAi of ADAMTS4 and -5 would be necessary to confirm contribution of specific ADAMTS members. However, in this publication we showed that neuronal activity induces a proteolytic cascade necessary for structural plasticity. The identification of exact proteases involved in this process is certainly the next logical step to go, but beyond the scope of the current study.

Additionally, concerns regarding the substantial variation in dendritic spine density and morphology persist. While it is noted that small p-values were obtained through careful planning and analysis (Figure 7F and G), the changes in the values of dendritic protrusion length and density between control and treated groups remain very small and subtle. I appreciate the authors' intention to reduce the number of experimental animals to be used for additional experiments in the revised submission. Comparing the same regions of dendrites in hippocampal slices before and after treatment would provide more compelling evidence without requiring a large sample size.

We carefully examined published data and found that the mean value and SEM values that we report are very similar to the values obtained by live imaging by others. We counted an average of 13.63 ± 0.66 spines per $10 \mu\text{m}$ (mean \pm SEM) in control slices. A study carried out in the Bonhoeffer laboratory (Scheuss & Bonhoeffer, 2014) using animals of a similar age (P70–90) and live imaging in acute slices found a spine density of 12.7 ± 0.6 spines/ $10 \mu\text{m}$ (mean \pm SEM) in CA1 pyramidal neurons. Another study that used organotypic slice cultures and live imaging counted 16 ± 0.5 spines/ $10 \mu\text{m}$ (Michaluk et al, 2011). Since the variations in the number of spines are very similar in all these examples, we conclude that the observed variation in spine number is based on naturally occurring biological variation rather than technical bias. Thus, live imaging of slices would not improve this situation, as it would encounter the same variability in spine density as in our study.

We report the average size of dendritic protrusions (spines) $0.76 \pm 0.027 \mu\text{m}$ in control slices. For comparison, Arellano et al. found spines to be, on average, $0.66 \pm 0.035 \mu\text{m}$, and Attardo et al. reported similar data, with spine lengths ranging from $0.2\text{--}1.5 \mu\text{m}$ (average $0.8 \mu\text{m}$) (Arellano et al, 2007; Attardo et al, 2015). Both studies reported similar average lengths and SEM to those observed in our study. Thus, we think that also here, the variability in length of dendritic protrusion represents the biological variability and not an inappropriate methodology.

Further, the reviewer claims that the reported changes are “very small and subtle”. The spine lengths measured in our study are as follows: Ctrl: 0.76 ± 0.027 ; PFR: 0.97 ± 0.021 ; and PFR+TIMP3: 0.77 ± 0.056 . These represent an PFR-induced increase in spine length by 31% compared to control and 27% compared to PFR with protease blocker TIMP3. The testing of statistical significance of these effects resulted in $p < 0.001$ (Šídák's multiple comparisons test). These changes are also biologically significant, since other studies that investigated the impact of altered neuronal activity on spine size and density reported similar extent of changes (Kirov & Harris, 1999; Maletic-Savatic et al, 1999; Shao et al, 2021). Thus, we cannot agree with the statement that the observed effects are very small and subtle.

Thus, since our approach revealed robust, biologically relevant and statistically highly significant changes, we consider it appropriate and sufficient for the aims of this study. In support of this, many studies employed the same methodology to test effects of specific

treatments on the number of dendritic protrusion and size (Kirov & Harris, 1999; Matsumoto-Miyai et al, 2009; Michaluk et al., 2011; Shao et al., 2021). Of course, if the focus is on the dynamic changes of dendritic spines, such as growth and shrinkage or fluctuations in spine numbers, the proposed experiments of reviewer 3 would be the method of choice. However, this was not our intention in this study. We also agree that taking snapshots of the same dendrite over the course of the experiment might require fewer animals than our experimental setup, since paired statistical analysis could be applied instead of the more stringent unpaired analysis. However, as previously mentioned, the variability in spine number and length would not change, nor would the effect size. Thus, the proposed experiments would not deliver any new insights but would represent a repetition of a previous experiment. In our view, this does not justify the use of additional animals, as no meaningful scientific gain is expected.

Literature:

Arellano JI, Benavides-Piccione R, Defelipe J, Yuste R (2007) Ultrastructure of dendritic spines: correlation between synaptic and spine morphologies. *Front Neurosci* 1: 131–143

Attardo A, Fitzgerald JE, Schnitzer MJ (2015) Impermanence of dendritic spines in live adult CA1 hippocampus. *Nature* 523: 592–596

Kirov SA, Harris KM (1999) Dendrites are more spiny on mature hippocampal neurons when synapses are inactivated. *Nat Neurosci* 2: 878–883

Maletic-Savatic M, Malinow R, Svoboda K (1999) Rapid dendritic morphogenesis in CA1 hippocampal dendrites induced by synaptic activity. *Science* 283: 1923–1927

Matsumoto-Miyai K, Sokolowska E, Zurlinden A, Gee CE, Luscher D, Hettwer S, Wolfel J, Ladner AP, Ster J, Gerber U et al (2009) Coincident pre- and postsynaptic activation induces dendritic filopodia via neurotrypsin-dependent agrin cleavage. *Cell* 136: 1161–1171

Michaluk P, Wawrzyniak M, Alot P, Szczot M, Wyrembek P, Mercik K, Medvedev N, Wilczek E, De Roo M, Zuschratter W et al (2011) Influence of matrix metalloproteinase MMP-9 on dendritic spine morphology. *J Cell Sci* 124: 3369–3380

Scheuss V, Bonhoeffer T (2014) Function of dendritic spines on hippocampal inhibitory neurons. *Cereb Cortex* 24: 3142–3153

Shao LX, Liao C, Gregg I, Davoudian PA, Savalia NK, Delagarza K, Kwan AC (2021) Psilocybin induces rapid and persistent growth of dendritic spines in frontal cortex in vivo. *Neuron* 109: 2535–2544 e2534

Dr. Renato Frischknecht
University of Erlangen-Nuremberg
Animal Physiology
Staudtstrasse 5
Erlangen, Bayern 91058
Germany

Dear Renato,

I am very pleased to accept your manuscript for publication in the next available issue of EMBO reports. Thank you for your contribution to our journal.
